

# An anticlockwise metamorphic P-T path and nappe stacking in the Reisa Nappe Complex in the Scandinavian Caledonides, northern Norway: evidence for weakening of lower continental crust before and during continental collision

Carly Faber[1], Holger Stünitz[1,2], Deta Gasser[3,4], Petr Jeřábek[5], Katrin Kraus[1], Fernando Corfu[6], Erling K. Ravna[1]; Jiří Konopásek[1]

[1]Department of Geosciences, UiT The Arctic University of Norway, Tromsø, N-9037, Norway
[2]Institut des Sciences de la Terre (ISTO), Université d´Orleans, 45100, France
[3]Western Norway University of Applied Sciences, Sogndal 6851, Norway
[4]Geological Survey of Norway, Trondheim 7491, Norway
[5]IPSG, Faculty of Science, Charles University, Albertov 6, 128 43, Prague 2, Czech Republic
[6]Department of Geosciences & Centre for Earth Evolution and Dynamics, University of Oslo, Norway

*Correspondence to*: Carly Faber (carlyfaber1@gmail.com)

**Abstract.** This study investigates the Caledonian metamorphic and tectonic evolution in northern Norway, examining the
structure and tectonostratigraphy of the Reisa Nappe Complex (RNC; from bottom to top, Vaddas, Kåfjord and Nordmannvik nappes).  Structural data, phase equilibrium modelling, and U-Pb zircon and titanite geochronology are used to constrain the timing and P-T conditions of deformation and metamorphism that formed the nappes and facilitated crustal thickening during continental collision. Five samples taken from different parts of the RNC reveal an anticlockwise P-T path attributed to the effects of early Silurian heating followed by thrusting. An early Caledonian S1 foliation in the Nordmannvik Nappe records
kyanite-grade partial melting at ~760 – 790 °C and ~9.4 – 11 kbar. Leucosomes formed at 439 ± 2 Ma (U-Pb zircon) in fold axial planes in the Nordmannvik Nappe indicate that compressional deformation initiated while the rocks were still partially molten. This stage was followed by pervasive solid-state shearing as the rocks cooled and solidified, forming the S2 foliation at 680 – 730 °C and 9.5 – 10.9 kbar. Multistage titanite growth in the Nordmannvik Nappe records this extended metamorphism between 444 and 427 Ma. In the underlying Kåfjord Nappe, garnet cores record lower P-T (590 – 610 °C and 5.5-6.8 kbar)
but a similar geothermal gradient as the S1 migmatitic event in the Nordmannvik Nappe, indicating formation at a higher relative position in the crust. S2 shearing in the Kåfjord Nappe occurred at 580-605 °C and 9.2-10.1 kbar, indicating a considerable pressure increase during nappe stacking. Gabbro intruded in the Vaddas Nappe at 439 ± 1 Ma, synchronously with migmatization in the Nordmannvik Nappe. In the Vaddas Nappe S2 shearing occurred at 630-640 ºC and 11.7-13 kbar. Titanite growth along the lower RNC boundary records S2-shearing at 432 ± 6 Ma. It emerges that early Silurian heating (~440
Ma), probably resulting from large-scale magma underplating, initiated partial melting that weakened the lower crust, which facilitated dismembering of the crust into individual nappe units.  This tectonic style contrasts subduction of mechanically strong continental crust to great depths.



## 1 Introduction

Continental collision is one of the most important processes in plate tectonics. Large-scale thrusting and nappe stacking are the main processes responsible for crustal shortening during continental collision. Depending on their rheology crustal rocks can be subducted to depths of ~200 km (e.g. Chopin, 2003; Spengler et al., 2009; Hacker et al., 2010), or be included in large-

scale nappe stacks (e.g. Escher et al., 1993; Escher and Beaumont, 1997). Crucial factors that control the style of large-scale deformation and mid- to lower crustal ductile nappe stacking are temperature, availability of fluids, composition of the rocks involved, and the presence or absence of melt (e.g. Hollister and Crawford, 1986; Beaumont et al., 2006; Gerya and Meilick, 2010; Labrousse et al., 2010; Jeřábek et al., 2012). The immediate pre-collisional history of rocks involved in nappe-stacking plays a significant role in determining these factors, and variations in pre-collisional history along strike affect the nature of

collisional processes along an orogen. In active continental collision zones such as the Himalaya, mid- and lower-crustal processes are generally inferred from surface deformation, geophysical information, and the geochemistry of erupted volcanic rocks (e.g. Schulte-Pelkum et al., 2005; Zhao et al., 2009). In contrast, ancient and deeply eroded continental collision zones such as the Caledonides allow for direct study of nappe thrusting at mid- and lower crustal levels, allowing insight into the processes of large-scale continental subduction and crustal shortening.

The Caledonides were formed by convergence and collision between Baltica and Laurentia in the Silurian-Devonian (Fig. 1A). The resulting large-scale nappe stacks provide access to the study of mid- to lower-crustal processes during a continental collision of Himalayan style and extent (e.g. Streule et al., 2010; Labrousse et al., 2010). In northern Norway, a well preserved section from the autochthonous Baltica basement to exotic terranes and ophiolites is exposed (Fig. 1B), displaying large

gradients in metamorphic grade and deformational style (Corfu et al., 2014). The Reisa Nappe Complex (RNC) represents a large ductile nappe stack, metamorphosed at amphibolite- to granulite-facies conditions, displaying pervasive ductile deformation and possible Caledonian partial melting (Roberts and Sturt, 1980; Andresen, 1988). Based on its similarity to rocks at the same tectonostratigraphic level (e.g. the Magerøy Nappe; Andersen, 1982, 1988; Corfu et al., 2006), the RNC is considered equivalent to other Iapetus-derived or outer Baltica margin (upper allochthon) rocks in the Caledonides. It may also

preserve an early Silurian history, recording events immediately prior to or during early continental collision between Baltica and Laurentia (e.g. Andréasson et al., 2003; Slagstad and Kirkland, 2018). In addition, its tectonostratigraphic position directly below the ophiolitic rocks of the Lyngsfjellet Nappe is paleogeographically unique. Understanding the structure and composition of the RNC together with its metamorphic, deformation, and magmatic history will help to establish how lower crustal nappe stacking during continental collision takes effect. The pre-Caledonian and Caledonian evolution of the rocks

affected the deformation behaviour during nappe stacking in this particular part of the Caledonian Orogen.



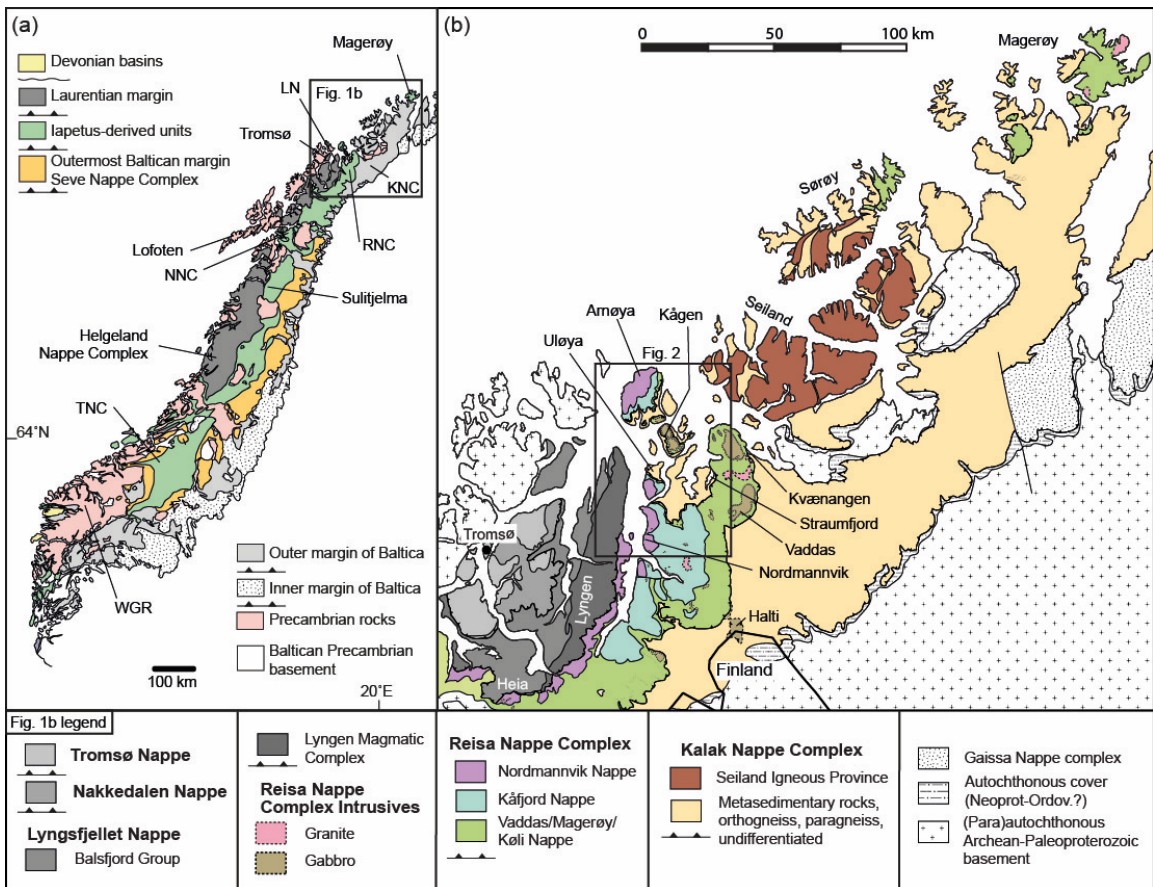

**Figure 1: (a) Simplified map of the Scandinavian Caledonides and their inferred palaeotectonic origin, modified from Gee et al. (2010). LN = Lyngsfjellet Nappe, WGR = Western Gneiss Region, NNC = Narvik Nappe Complex, KNC = Kalak Nappe Complex, RNC = Reisa Nappe Complex., TNC = Trondheim Nappe Complex. (b) Map showing the extent of the Reisa Nappe Complex in northern Norway, based on Zwaan (1988) and own correlations. The Vaddas, Kåfjord and Nordmannvik nappes are shown in colour. The study area is denoted by the black box (Fig. 2).**

## 2 Geological Framework

The Scandinavian Caledonides consist of a series of allochthonous nappes, with from bottom to top Baltican, Iapetus, or Laurentian affinities (Roberts and Gee, 1985; Stephens & Gee, 1989; Fig. 1A).





### 2.1 The North Norwegian Caledonides

In northern Norway a well-preserved section through the Caledonian nappe stack is exposed (Fig. 1B; e.g. Ramsay et al., 1985). The Baltican basement with its metasedimentary autochthonous cover is overlain by the parautochthonous metasedimentary rocks of the Gaissa Nappe Complex. The allochtonous rocks form a nappe stack including the following

units, from bottom to top: 1) the metasedimentary and metaigneous rocks of the Kalak Nappe Complex (KNC), 2) the metasedimentary and metaigneous rocks of the Reisa Nappe Complex (RNC), 3) the ophiolitic and metasedimentary rocks of the Lyngsfjellet Nappe, 4) gneisses and migmatites of the Nakkedalen Nappe, and 5) metasedimentary rocks of the Tromsø Nappe (Fig. 1B; Andresen, 1988; Kvassnes et al., 2004; Kirkland et al., 2006, 2007a Augland et al., 2014; Rice, 2014; Gee et al., 2017). Caledonian metamorphic conditions associated with nappe thrusting generally increase upwards through the KNC

and RNC, from low greenschist facies in the parautochtonous metasediments to upper amphibolite/lower granulite facies at the top of the RNC, with local granulite facies metamorphism generally interpreted to be pre-Caledonian (Andresen, 1988; Elvevold et al., 1993, 1994;). The Lyngsfjellet Nappe displays greenschist facies metamorphism at its base with higher-grade metamorphism in the overlying metasediments, and the overlying Nakkedalen and Tromsø nappes show amphibolite to eclogite facies metamorphism (Andresen and Bergh, 1985; Andresen & Steltenpohl, 1994; Corfu et al., 2003). The main

Caledonian deformation (often referred to as Scandian; e.g. Corfu et al., 2014) in the allochthons in northern Norway is associated with top-to-SE and top-to-E shearing (Rice, 1998).

The palaeogeographic origin of several of the Caledonian nappes in northern Norway is debated. The Gaissa Nappe Complex is interpreted to represent telescoped Baltica margin cover, and the KNC has traditionally also been interpreted to represent

more outboard Baltica basement and its late Precambrian to Paleozoic cover (e.g. Stephens and Gee, 1989; Gee et al., 2017). However, recent evidence shows that the sedimentary cover had already been deposited between 1000 and 900 Ma, and the KNC underwent metamorphism, deformation and magmatism during several events at about 970-950 Ma, 870-830 Ma, 700 Ma, 600 Ma and 570 Ma. These Neoproterozoic events are not known in the Baltica basement elsewhere, implying that either the KNC has an exotic origin or that a still unknown type of suture (typical suture rocks are lacking beneath the KNC) separated

it from the Archean – Palaeoproterozoic northern Baltic basement before the final collision (e.g. Kirkland et al., 2006, 2007a, 2007, 2008; Corfu et al., 2007, 2011). The RNC has previously been interpreted to represent either Iapetus-derived or outermost Baltican rocks, whereas the Lyngsfjellet Nappe is traditionally considered as Iapetus-derived, marking the transition towards the Laurentia-derived Nakkedalen and Tromsø Nappes (e.g. Stephens and Gee, 1989; Corfu et al., 2003).

### 2.2 The Reisa Nappe Complex (RNC)

The RNC outcrops east of Lyngen (Fig. 1B) and includes, from bottom to top, the Vaddas, Kåfjord and Nordmannvik nappes (Figs. 1, 2; Zwaan & Roberts, 1978; Zwaan, 1988). The Vaddas Nappe has been divided into lower and upper parts (Fig. 3; Lindahl et al., 2005). The lower part (lower Vaddas) is mapped only between Straumfjord and Kvænangen and includes a






**Figure 2: Structural map, stereonets and cross sections from the study area. Petrological and geochronological samples locations are indicated. Structures are shown on equal area lower hemisphere stereographic projections as follows: (a) the RNC on northern and western Arnøya, (b) the RNC on southernmost Arnøya, (c) the KNC on Uløya; poles to S2 plot on a great circle (dashed line), the pole of which (black box) defines a fold hinge parallel to plotted stretching lineations, (d) the RNC on Uløya, (e) the Nordmannvik Nappe on the eastern coast of Lyngen, and (f) the Vaddas Nappe on the eastern side of Reisafjord; poles to foliation define a folding event with the fold hinge (black box) parallel to the L2 stretching lineation and measured open fold axes. Cross sections are shown at the bottom of the figure from Arnøya to Eide (A-B) and from Lyngen to Sørkjosen (C-D). The density of red lines corresponds to the intensity of ductile deformation within and along boundaries between individual nappes. Caledonian foliations are shown in red, whereas black are possibly older.**

basal marble and calc-silicate layer, garnet- and graphite-bearing schist, meta-arkose, quartzite and amphibolite, representing metamorphosed volcanosedimentary rocks and turbidites (Andresen, 1988; Lindahl et al., 2005). The 602 ± 5 Ma Rappesvarre metagranite occurs in the lower Vaddas Nappe, although its relationship with the surrounding rocks is unclear. It appears to be equivalent to rocks in the Corrovarre Nappe, at the top of the KNC (Figs. 2, 3; Lindahl et al., 2005; Corfu et al., 2007; Gee et al., 2017).

The base of the upper part of the Vaddas Nappe (upper Vaddas) includes a quartzite-conglomerate and marble layer with local late Ordovician-early Silurian fossils (447-441 Ma; Fig. 3; Binns and Gayer, 1980). Above, garnet-bearing calc-schist with layers and lenses of amphibolite underlie a thick sequence of metapsammites (Zwaan, 1988; Lindahl et al., 2005). Several undated tholeiitic gabbro bodies intrude the upper Vaddas rocks (Kågen, Kvænangen and Vaddas gabbros, Figs. 2, 3; Lindahl et al., 2005). The Halti Igneous Complex (Fig. 1B), a klippe of the Vaddas Nappe (Vaasjoki and Sipilä, 2001), but also interpreted as an intrusion in the KNC (Andréasson et al., 2003), yields intrusion ages of 434 ± 5 Ma and 438 ± 5 Ma. The Magerøy Nappe and Hellefjord schists in the northeast and Køli Nappe in the southwest are thought to be equivalents of the Vaddas Nappe (Fig. 1B; Gayer & Roberts, 1973; Andersen, 1981; Lindahl et al., 2005; Corfu et al., 2006, 2011; Kirkland et al., 2005, 2016). Volcaniclastic and intrusive rocks in the Magerøy Nappe give early Silurian ages (442-435 Ma) suggesting almost synchronous deposition and intrusion (Robins, 1998; Corfu et al., 2006, 2011; Kirkland et al., 2005, 2016).

Both the Vaddas and Kåfjord nappes display amphibolite facies metamorphic conditions and pervasive shearing. They are separated by a mylonite zone (Zwaan and Roberts, 1978; Andresen, 1988; Zwaan, 1988). The lower part of the Kåfjord Nappe is composed of marble and calc-schist, metapsammite and garnet mica-schist. Mylonitic gneisses with boudinaged amphibolite and granitic bodies dominate the upper part of the nappe (Andresen, 1988). Small gabbro bodies of unknown age also occur (Zwaan, 1988). A Rb-Sr whole-rock age from the upper part of the nappe suggests anatexis and granite crystallization at 440 Ma (Dangla et al., 1978). The boundary between the Kåfjord and the overlying Nordmannvik Nappe is a well-developed mylonitic zone (Andresen, 1988). The Nordmannvik Nappe, defined east of Lyngen at Nordmannvik and best studied from Nordmannvik to Heia (Fig. 1B), is a polymetamorphic nappe showing a pervasive amphibolite facies foliation surrounding granulite facies relict lenses (Elvevold, 1987; Andresen, 1988; Zwaan, 1988; Lindstrøm and Andresen, 1992; Augland et al., 2014). The nappe is comprised of garnet-mica-schist and gneiss, migmatite, minor calc-silicate, amphibolite, and marble (Fig.




3). Small bodies of gabbro and sagvandite (metasomatic carbonate-orthopyroxenite) occur within the nappe (Schreyer et al., 1972; Lindstrøm and Andresen, 1992). Granulite facies metamorphic conditions at Heia (Fig. 1B) were estimated at $715 \pm 30$ °C and $9.2 \pm 1$ kbar using multiple geothermometers and geobarometers (Elvevold, 1987). U-Pb zircon ages from the rocks at Heia indicate high-temperature metamorphism at $439 \pm 1$ Ma and gabbro crystallization at $435 \pm 1$ Ma, overprinted by

Caledonian shearing at $420 \pm 4$ Ma (Augland et al., 2014). Greenschist facies mylonitic rocks mark the boundary between the Nordmannvik Nappe and Lyngsfjellet Nappe above. The Lyngsfjellet Nappe is comprised of greenschist to amphibolite facies fossiliferous metasedimentary rocks unconformably overlying the mafic-ultramafic Lyngen Magmatic Complex, interpreted as an ophiolite formed in an incipient arc setting in the Early Ordovician (possible Laurentian arc; Figs. 1, 3; Andresen and Bergh, 1985; Zwaan, 1988; Stephens and Gee, 1989; Andresen and Steltenpohl, 1994; Kvassnes et al., 2004; Augland et al.,

2014). Timing and kinematics of emplacement of the Lyngsfjellet Nappe over the Nordmannvik Nappe are unclear.

### 3 Field Results

We investigated the RNC east and north of Lyngen in coastal areas, where it is exposed around a window of KNC rocks (Figs. 1, 2). Detailed structural sections through the RNC were investigated on Uløya, supplemented by structural mapping along the east coast of Lyngen, on Kågen and around Straumfjord, and through the nappe stack on Arnøya (Figs. 2, 3). The nappes on

Arnøya were previously assigned to the KNC (Roberts, 1973; Zwaan, 1988), but later reconsidered as being part of the Vaddas Nappe (Andresen, 1988). More recent maps have classified northern Arnøya as part of the Magerøy/Vaddas Nappe and southern Arnøya as KNC (e.g. Corfu et al., 2007; Augland et al., 2014; Gasser et al., 2015). Our own field observations indicate that all three RNC nappes (Vaddas, Kåfjord and Nordmannvik) are present on Arnøya (Figs. 1B, 2, 3).

### 3.1 Lithologies

The lower boundary of the RNC towards the KNC was investigated on Uløya and Arnøya (Fig. 2). The underlying KNC comprises either quartzofeldspathic rocks or metapelitic paragneisses (Figs. 3, 4A), whereas the boundary itself consists of a ~40 m-thick strongly mylonitized zone comprising intercalated amphibolite facies quartzofeldspathic rocks, gneisses, and schists (Fig. 4B). Marbles are locally present together with the amphibolite and schist. On Uløya a migmatitic paragneiss that forms part of the KNC below the boundary is subsequently mylonitized in the solid state, with cm-sized porphyroclasts of

garnet and feldspar (Fig. 4B, C). Kinematic indicators consistently show top-to-SE shearing. Above the mylonitic layer, sheared metaconglomerate (Fig. 4D) occurs together with a marble unit, which marks the base of the Vaddas Nappe on Arnøya, Kågen and Uløya (Fig. 3). This marble layer is overlain by amphibolite and calc-schist (Fig. 4E). Two main marble units are found in the Vaddas Nappe at Straumfjord; one in the lower Vaddas and one in the upper Vaddas (Fig. 3). The marble at the base of the lower Vaddas is variably sheared and overlain by intercalated metasediments and metapsammites similar to those

in the KNC. The upper marble layer marks the base of the upper Vaddas and is associated with sheared conglomerates. It is overlain by amphibolite and calc-schist, similar to the Vaddas succession on Arnøya, Kågen, and Uløya, and we consider that




only the upper Vaddas is found on these three islands (Fig. 3). On Arnøya, Kågen and in Straumfjord, gabbro intrusions in the upper Vaddas Nappe (Figs. 2, 3) are associated with local migmatization of the surrounding metasediments, showing a clear intrusive relationship.

A well-developed strongly mylonitic foliation in muscovite-rich garnet-mica-schist marks the Vaddas-Kåfjord nappe boundary on Uløya and Arnøya (Figs. 2, 3). The boundary cuts a gabbro body in the Vaddas Nappe on Arnøya (Fig. 3). The Kåfjord Nappe is comprised of a lower unit of homogenous garnet-mica-zoisite-schist with calc-silicate lenses and some amphibolite layers (Fig. 4G-I), and a homogenous upper unit comprised of garnet-mica-schist and gneiss. This upper unit often displays strongly sheared layers rich in quartz-feldspar sigma clasts, possibly indicating the presence of leucosome prior to
mylonitization. The lower Kåfjord is similar to the calc-schists in the underlying Vaddas Nappe, whereas the upper Kåfjord is similar to strongly sheared rocks in the Nordmannvik Nappe (e.g. Figs. 3, 5A). Both units are pervasively and strongly sheared at amphibolite facies conditions. The Kåfjord-Nordmannvik boundary consists of mylonitic gneisses and garnet-mica-schist at least 50 m-thick. The Nordmannvik Nappe is mainly comprised of garnet-biotite-gneiss with layers and lenses of amphibolite and local calc-silicates, generally dominated by a mylonitic foliation (Fig. 5A, 5B).  Further away from the nappe boundary
rare relict lenses (< 50 m) display a higher-grade migmatitic foliation (Fig. 5C). On Arnøya, the frequency of relict lenses increases upwards away from the nappe boundary. The migmatite comprises felsic leucosome and biotite, garnet, and kyanite in the restite. The amount of leucosome varies spatially between 5-25% (Figs. 5D, E). Layers of amphibolite in the nappe also contain minor tonalitic leucosome suggesting that the mafic rocks have also been migmatized.

### 3.2 Structures

The earliest observed structural element is the syn-migmatitic foliation observed in the low strain lenses in the Nordmannvik Nappe (S1; Figs. 2A, 2E, 5C-E). In most cases it is sheared and overprinted by the solid-state amphibolite-facies S2 mylonitic foliation, which is pervasive throughout all the nappes of the RNC and underlying KNC (Fig. 2A-F). In the lower strain lenses the S1 migmatitic foliation is found in 3 structural orientations: 1) as a steep foliation that shows a variable trend (rare; Figs. 5C), 2) folded in open to closed folds with axial planes (containing leucosome) parallel to the S2 foliation (Fig. 5D), and 3)
most commonly as a shallowly-dipping foliation, parallel to the S2 foliation (Fig. 2A, D, E). Fold hinges in the migmatites (F2) plunge towards SE or NW, parallel to the L2 stretching lineation. An L2 intersection lineation between S1 and S2 is observed in the Nordmannvik and upper Kåfjord nappes. On Arnøya it is mostly parallel to the L2 stretching lineation, whereas on Uløya it also plunges shallowly towards the SW and W as well, consistent with the presence of a variable S1 foliation in the upper Kåfjord Nappe (Fig 2A, D, E).

The amphibolite-facies solid-state S2 foliation is generally flat-lying, dipping slightly to the west/north-west in the northern and western parts of the area and toward the east and southeast in the eastern part of the field area. The main variation is around the larger gabbro bodies, where S2 is deflected around them (Fig. 2). Gabbro interiors are nearly unaffected by S2 shearing,



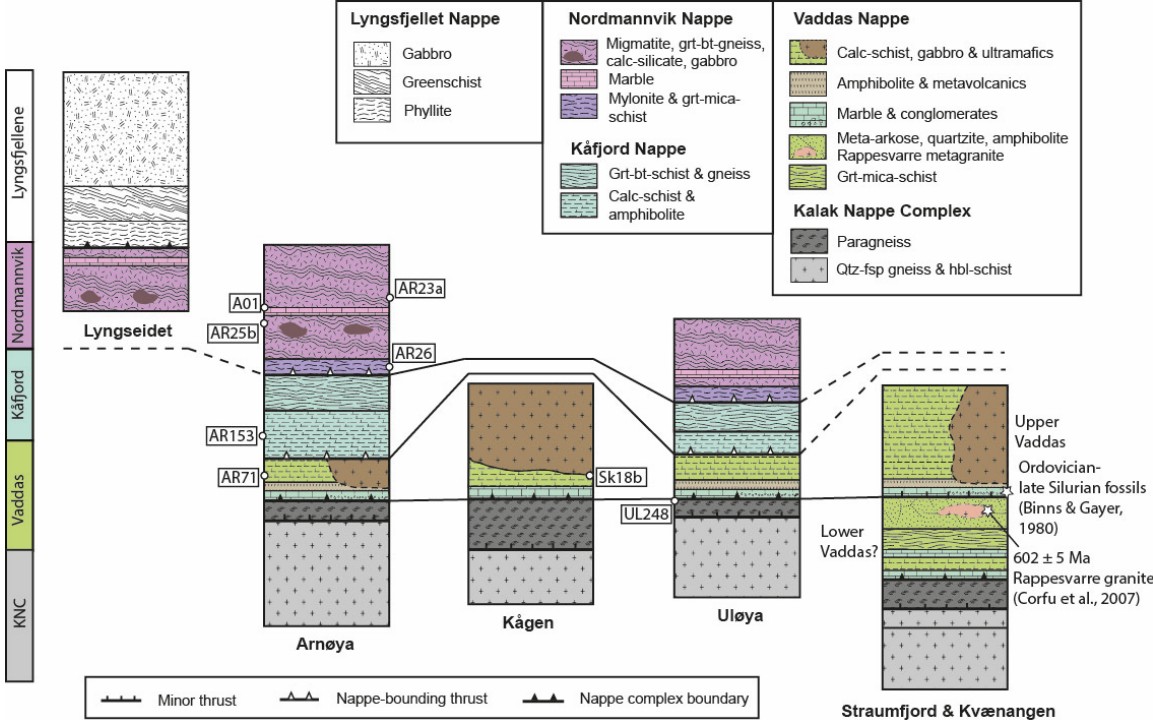

**Figure 3: Representative tectonostratigraphy of the RNC from Lyngseidet (west) to Straumfjord and Kvænangen (east) showing the location of major structures and relative thickness and spatial variation of tectonic units. Based on Zwaan (1988), Lindahl et al., (2005) and own work. Tectonostratigraphic position of the investigated samples is indicated.**

with most of the deformation confined along their boundaries. The S2 foliation is associated with a generally NW-SE-trending stretching lineation (L2; Fig. 2) and top-to-SE shear sense indicators (Fig. 4B, F, G). In the Vaddas and lower Kåfjord metasediments local micro- to meso-scale, isoclinal folds transpose a pre-S2 fabric. Dismembered fold hinges are common, with fold axes normally parallel to the L2 stretching lineation and axial planes parallel to S2 (Fig. 2A, B, D, E). The KNC-Vaddas, Vaddas-Kåfjord and Kåfjord-Nordmannvik nappe boundaries generally display a relatively stronger S2 mylonitic foliation than within the nappes, and the L2 stretching lineation at the nappe boundaries is often more pronounced than within nappe interiors.



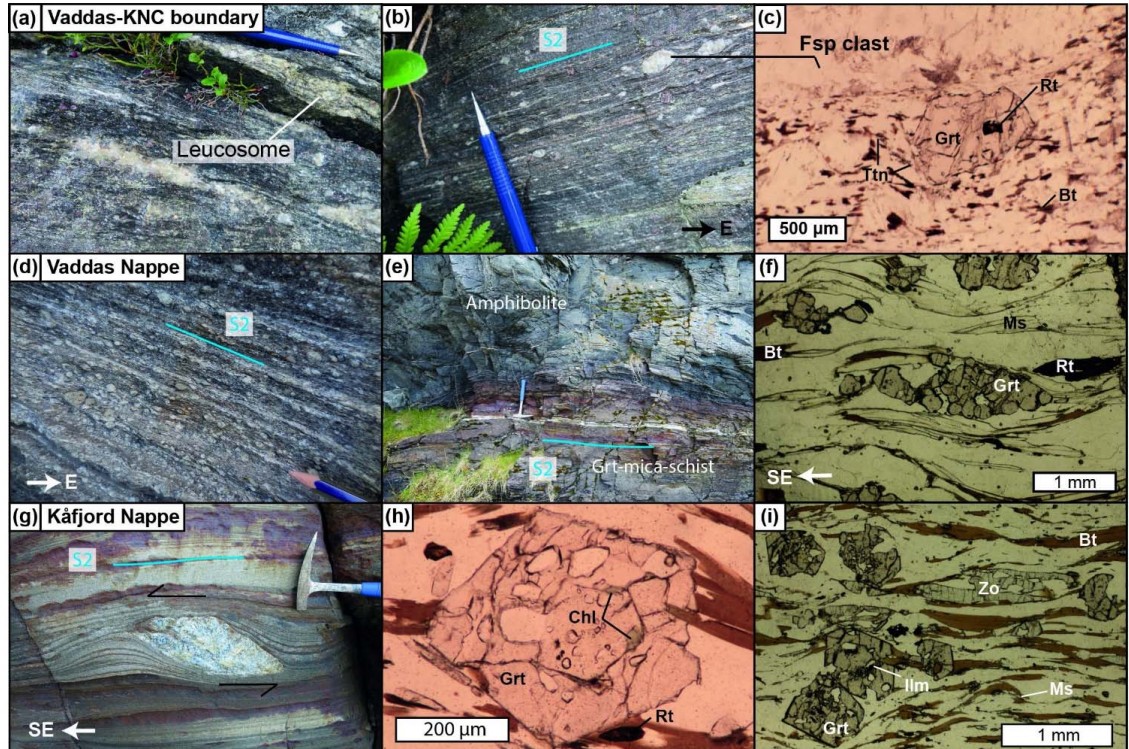

**Figure 4: Field photographs and photomicrographs from Vaddas and Kåfjord nappes. Coordinates for sample sites are given in Table 1. (a) KNC paragneiss with visible (sheared) leucosome from 30 m below the Vaddas-KNC boundary shear zone at site UL248. (b) The Vaddas-KNC boundary mylonite at site UL248. (c) Photomicrograph of sample UL248 showing the edge of a K-feldspar porphyroclast, garnet with a rutile inclusion, and matrix with biotite and titanite visible. (d) Sheared metaconglomerate from the base of the Vaddas Nappe on Uløya at N69.89326°, E20.55713° with mostly rounded quartzite clasts in a pelitic matrix of biotite and garnet. (e) Typical strongly sheared garnet-mica-schist and amphibolite in the Vaddas Nappe at N69.89326°, E20.55456° (Uløya). (f) Photomicrograph of Vaddas Nappe sample AR71 showing sheared garnet (top-to-SE shear sense), and muscovite fish, biotite and rutile in the matrix. (g) A sheared calc-silicate lens (top-to-SE shear sense) in typical garnet-zoisite-biotite-schist from the lower Kåfjord Nappe at N69.85701°, E20.53317°. (h) Photomicrograph of garnet in Kåfjord Nappe sample AR153 showing distinctive core and rim structure with chlorite inclusions in garnet cores and rutile in the matrix. (i) Photomicrograph of Kåfjord Nappe sample AR153 showing garnet porphyblasts in a matrix with S2 foliation defined by zoisite, biotite and muscovite. An ilmenite inclusion in garnet is also shown.**





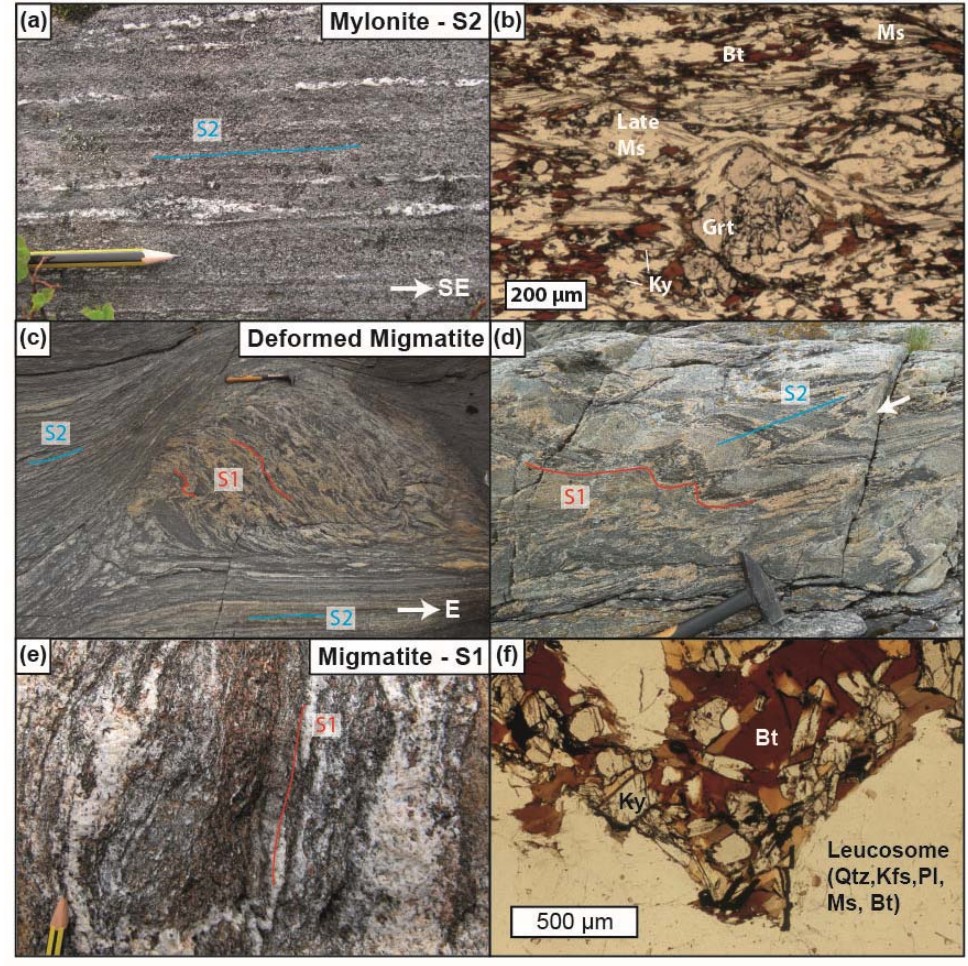

**Figure 5: Field photographs and photomicrographs from the Nordmannvik Nappe. (a)** Nordmannvik mylonite (solid-state deformed (S2) migmatite) at sample site AR26 near the Nordmannvik-Kåfjord boundary. **(b)** Photomicrograph of sample AR26 showing garnet porphyroblasts with sillimanite inclusions in a matrix and a strong S2 foliation defined by biotite, kyanite and muscovite. **(c)** A lens-shaped domain in the Nordmannvik Nappe on Uløya at N69.84046°, E20.51852° with an older S1 migmatitic foliation overprinted and sheared by solid-state S2 mylonitic foliation. **(d)** Nordmannvik migmatite at site AR23 showing the folded S1 migmatitic foliation (Caledonian fold geometry) with melt segregations in an axial planar orientation (white arrow; parallel to S2). **(e)** Nordmannvik migmatite at sample site AR25b showing a high volume of leucosome and garnet- and biotite-rich melanosomes. **(f)** Photomicrograph of sample AR25b showing a lack of S2 foliation, and kyanite and biotite along the edge of a leucosome segregation.





### 3.3 Tectonostratigraphic interpretation

Based on our field observations we propose a correlation of the different parts of the RNC across the study area (Fig. 3). Lithological similarities indicate that the basal marble/conglomerate layer of the Vaddas Nappe on Arnøya, Kågen, and Uløya correlates with the uppermost marble/conglomerate layer at the base of the upper Vaddas Nappe around Straumfjord at a

similar tectonostratigraphic level to where Binns and Gayer (1980) identified early Silurian fossils (Fig. 3). This indicates that the lower Vaddas rocks, which are relatively thick in the eastern study area, thin out towards the north and west and are not present elsewhere in the field area (Fig. 3). The lower Vaddas unit includes the Rappesvarre granite, and, based on the structural and lithological similarity to the underlying KNC, we favour the interpretation that it is either a composite nappe of KNC and upper Vaddas rocks, or re-worked KNC rocks, as suggested by Corfu et al., (2007) and Gee et al., (2017). More work needs

to be done to establish whether it has more RNC or KNC affinity (perhaps Corrovarre Nappe) to constrain its extent south of the field area.

Gabbros are characteristic of the upper Vaddas Nappe, and the continuation of the Kågen gabbro on southern Arnøya indicates that the upper Vaddas Nappe is present there as well (Figs. 2, 3). Our investigation on Arnøya shows an identical nappe stack

as present on Uløya, indicating that the northwestern half of Arnøya is part of the RNC, including the upper Vaddas, Kåfjord and Nordmannvik nappes (Figs. 2, 3). The kyanite-bearing migmatites and gneisses on the northern part of Arnøya are assigned to the Nordmannvik Nappe based on their similarity to migmatites at Lyngseidet and on southern Uløya, and their tectonostratigraphic position (Fig. 3). We separate the Kåfjord Nappe, based on lithological differences, into upper and lower parts. The upper part is similar to the strongly sheared rocks of the Nordmannvik Nappe, and records a Rb-Sr whole rock ~440

Ma age, possibly for anatexis (Dangla et al., 1978). It could alternatively be considered as part of the Nordmannvik Nappe. The lower part consists of calcareous metasediments that show no evidence for prior anatexis and are more similar to the underlying metasediments of the Vaddas Nappe. Since the lower boundary of the lower Kåfjord cuts the upper part of the gabbro in the Vaddas Nappe on Arnøya, it probably forms a thin thrust sheet of Vaddas-type metasediments overlying the Vaddas Nappe. All nappes show significant thickness variations throughout the study area (Fig. 3). The entire Vaddas Nappe,

as it is currently defined, shows thicknesses of 100-1500 m with its lower part wedging out towards the west and north. The Kåfjord Nappe is thickest in its southern part (>1000 m) and thins towards Arnøya (< 1000 m), whereas the Nordmannvik Nappe is much thicker (> 2000 m) than the Vaddas or Kåfjord nappes and thickest on Arnøya (Figs. 2, 3). This results in flat-lying laterally extensive but lensoid thrust sheets, which are separated by mylonitic shear zones but which are internally pervasively sheared (Fig. 2, sections A`-B` and C`-D`).




## 4 Metamorphism in the RNC

Metamorphic conditions were investigated throughout the entire RNC in order to resolve variations in P-T and deformation conditions throughout the nappe stack. Sample sites and details are shown in Figures 2, 3 and Table 1. Table 2 gives a summary of the mineralogy in the samples used for P-T modelling. Mineral abbreviations are according to Whitney and Evans (2010).

### 4.1 Methods

P–T conditions for all samples were estimated using phase equilibrium modelling with the Perple_X software (Connolly, 2005: version 6.6.6) using the internally consistent thermodynamic data set of Holland & Powell (1998: 2004 upgrade). The calculations were performed in the MnNCKFMASHTi system using XRF whole-rock compositions (Table S1). The following solution mixing models were used: garnet, staurolite, chloritoid (Holland & Powell, 1998), biotite (Tajčmanová et al., 2009),

ternary feldspar (Fuhrman & Lindsley, 1988), ilmenite (ideal mixing of ilmenite, geikielite, and pyrophanite end-members), melt (Holland & Powell, 2001) and white mica (Coggon & Holland, 2002). Iron was assumed to be $Fe^{2+}$ as the $Fe^{3+}$ content of the minerals considered is negligible and $Fe^{3+}$ oxides occur in negligible amounts. Apatite was observed in all samples and therefore the corresponding amount of CaO bonded to $P_2O_5$ observed in the whole rock analyses was subtracted from the bulk compositions. Measured chemical compositions of the relevant minerals (Tables 3 and 4) were compared with model isopleths

in the calculated pseudosections. Molar percent of grossular (Grs) and spessartine (Sps) end-members and the $X_{Mg}$ (Mg/Mg + $Fe_{tot}$) value in garnet are used to constrain P-T conditions, and are shown on the P-T sections. Estimated P-T conditions were checked with anorthite content in plagioclase (An = Ca/(Ca + Na + K)), $X_{Mg}$ (Mg/(Mg + Mn + $Fe_{tot}$)) in biotite, and Si content in white mica, and are shown on some pseudosections where they assist with further constraining P-T estimates. Fluid content was derived from the LOI value unless stated otherwise, and considered as pure $H_2O$ in all calculations.

| Sample | Nappe | Rock type | Sample site | Method | Context |
|--------|-------|-----------|-------------|--------|---------|
| UL248 | Kalak-Vaddas | Grt-bt-mylonite | N69.86755 °, E20.60304° | P-T modelling, U-Pb titanite | Lower part of ~40 m-thick mylonite zone |
| AR71 | Vaddas | Grt-micaschist | N70.06315°, E20.45522° | P-T modelling | Upper ~30 m of ~120m-thick Vaddas Nappe |
| Sk18b | Vaddas | Gabbroic pegmatite | N69.985900°, E20.829567° | U-Pb zircon | Pegmatite near the edge of Kågen gabbro |
| AR153 | Kåfjord | Grt-bt-zo-schist | N70.06797 °, E20.44341° | P-T modelling | Mid- to lower- Kåfjord Nappe |
| AR23a | Nordmannvik | Pelitic migmatite | N70.205343°, E20.522562° | U-Pb zircon | Lens with S1 foliation |
| AR25b | Nordmannvik | Pelitic migmatite | N70.16425°, E20.52112° | P-T modelling | Lens with S1 foliation |
| AR26 | Nordmannvik | Grt-bt-ky-schist | N70.137455°, E20.561118° | P-T modelling | Mylonite ~50 m above Nordmannvik-Kåfjord boundary (S2 foliation) |
| A01 | Nordmannvik | Calc-silicate | N70.191422°, E20.503611° | U-Pb titanite | S1 lens, interaction zone between calc-silicate layers and leucosome |

**Table 1: List of samples with rock types, sample sites and methods.**





### 4.2 Kalak-Vaddas boundary (UL248)

### 4.2.1 Petrography and mineral chemistry

Sample UL248 (Fig. 2, 3, 4B; Table 1) represents a mylonitized migmatitic paragneiss from the upper part of the KNC (Fig. 4A, 4B). The fine-grained matrix of sample UL248 contains porphyroblasts of garnet (up to 0.9 mm) and fragments (up to 5

5  mm) of plagioclase, K-feldspar and quartz-feldspar aggregates (Fig. 4C). Biotite, muscovite and elongate grains and aggregates of quartz and feldspar define the S2 mylonitic foliation. Two generations of muscovite are found as 1) rare large (0.2 mm) mica fish parallel to the S2 foliation (Ms1), and 2) as small grains intergrown with biotite and along garnet and K-feldspar boundaries, within the S2 foliation (Ms2). Quartz, biotite and rutile are common as inclusions in garnet (Fig. 4C). Minor rutile is also found in the matrix. Titanite is abundant as elongate 0.02-0.35 mm-long grains parallel to the S2 foliation and as

10  inclusions in garnet rims (Fig. 4C, 6A). Distinctive garnet zoning displays two generations of garnet. Garnet cores (Grt1) have a relatively flat compositional profile, followed by a transition zone to a ~0.1-0.15 mm-thick rim (Grt2; Fig. 6A). The two generations show significantly different compositions. Grt1 is lower in Grs ($Grs_{10-12}$), higher in spessartine ($Sps_{12-15}$), and has a higher $X_{Mg}$ content (0.11-0.13) than Grt2. Grt2 displays the following composition: $Grs_{36-38}$, $Sps_{3-4}$ and $X_{Mg}$ content of 0.08-0.11. Grt2 (rims) grew together with biotite, titanite, plagioclase (rims) and Ms2 during top-to-SE S2 shearing. $X_{Mg}$ in biotite

15  ranges between 0.36 and 0.45. Plagioclase generally shows zoning with a higher anorthite content in the cores ($An_{25-28}$) than in rims ($An_{20-23}$). Early muscovite (Ms1) has a Si content of 3.06-3.08 (a.p.f.u.) whereas later muscovite (Ms2) associated with S2 shearing has a higher Si content of 3.20-3.27 (a.p.f.u.; Table 4). The relict compositions shown by Grt1, plagioclase cores and large Ms1 fish indicate that pre-S2 P-T conditions may be preserved in this sample.

| Nappe | Sample | Main minerals | | | | | | | | | | Accessory minerals | | | | | | |
|---|---|---|---|---|---|---|---|---|---|---|---|---|---|---|---|---|---|---|
| | | Bt | Pl | Qtz | Grt | Kfs | Sil | Ky | Zo | Ms | Chl | Zrn | Mnz | Rt | Ttn | Ill | Ap | Ep/All |
| Kalak-Vaddas | UL248 | x | x | x | x | x | | | | x | | x | | x | x | x | x | x |
| Vaddas | AR71 | x | x | x | x | | | | | x | Gt in. | x | | x | | x | x | |
| Kåfjord | AR153 | x | x | x | x | | | x | x | x | | x | | x | x | x | x | |
| Nordmannvik | AR25b | x | x | x | x | x | x Gt in. | x | | x | | x | x | | | x | x | |
| Nordmannvik | AR26 | x | x | x | x | | x | x | | x | | x | | | x | x | | |

**Table 2: Mineralogy for petrology samples.**





| Mineral | Grt | | | | | | | | | | | | | | | | | | | |
|---|---|---|---|---|---|---|---|---|---|---|---|---|---|---|---|---|---|---|---|---|
| Nappe | Kalak-Vaddas Boundary | | | | Vaddas | | | | Kåfjord | | | | Nordmannvik (migmatite) | | | | Nordmannvik (mylonite) | | | |
| Sample | UL248 | | | | AR71 | | | | AR153 | | | | AR25b | | | | AR26 | | | |
| | Core (Grt1) | Rim (Grt2) | Core (Grt1) | Rim (Grt2) | Core | Rim (late) | Core | Rim | Core | Rim | Core | Rim | Core | Rim | Core | Rim | Grt core (lrg) | Grt rim (lrg) | Grt core (sml) | Grt rim (sml) |
| Wt% | | | | | | | * | * | | | | | * | * | * | * | * | * | * | * |
| $SiO_2$ | 37.63 | 38.01 | 37.47 | 38.00 | 37.74 | 37.76 | 37.72 | 37.58 | 37.81 | 37.60 | 37.60 | 37.59 | 38.61 | 38.14 | 37.87 | 37.59 | 37.76 | 37.67 | 37.72 | 37.91 |
| $TiO_2$ | 0.00 | 0.07 | 0.00 | 0.08 | 0.07 | 0.04 | 0.00 | 0.00 | 0.13 | 0.05 | 0.01 | 0.05 | 0.03 | 0.04 | 0.00 | 0.00 | 0.01 | 0.03 | 0.01 | 0.03 |
| $Al_2O_3$ | 21.64 | 21.41 | 21.49 | 21.46 | 21.81 | 21.82 | 22.00 | 21.77 | 21.94 | 22.51 | 22.05 | 22.57 | 22.00 | 21.92 | 21.89 | 21.64 | 21.67 | 21.60 | 21.76 | 21.56 |
| $Fe_2O_3$ | 0.00 | 0.63 | 0.00 | 0.00 | 0.00 | 0.00 | 0.00 | 0.00 | 0.00 | 0.00 | 0.00 | 0.00 | 0.00 | 0.00 | 0.00 | 0.00 | 0.00 | 0.00 | 0.00 | 0.00 |
| $FeO$ | 28.74 | 24.13 | 29.33 | 24.01 | 33.44 | 31.45 | 32.71 | 33.26 | 27.08 | 26.12 | 28.03 | 27.25 | 30.13 | 31.28 | 31.35 | 31.21 | 33.77 | 34.26 | 32.25 | 33.00 |
| $MnO$ | 6.25 | 1.66 | 6.06 | 1.76 | 0.69 | 0.63 | 0.64 | 0.70 | 1.54 | 0.80 | 1.55 | 0.86 | 1.44 | 1.61 | 1.62 | 1.63 | 0.49 | 0.52 | 1.53 | 1.58 |
| $MgO$ | 2.24 | 1.38 | 2.28 | 1.32 | 3.48 | 2.32 | 3.54 | 3.18 | 3.06 | 3.25 | 3.13 | 3.09 | 5.90 | 4.63 | 5.32 | 4.85 | 4.53 | 4.08 | 4.54 | 3.53 |
| $CaO$ | 4.52 | 13.46 | 4.02 | 13.38 | 2.73 | 5.35 | 2.62 | 3.19 | 7.91 | 8.79 | 6.89 | 8.04 | 1.70 | 2.07 | 2.06 | 2.50 | 1.56 | 1.68 | 2.11 | 2.18 |
| Total | 101.02 | 100.75 | 100.65 | 100.01 | 99.96 | 99.37 | 99.23 | 99.68 | 99.47 | 99.12 | 99.26 | 99.45 | 99.81 | 99.69 | 100.11 | 99.42 | 99.79 | 99.84 | 99.92 | 99.79 |
| $Si$ | 2.991 | 2.986 | 2.993 | 3.003 | 3.011 | 3.034 | 3.029 | 3.008 | 3.001 | 2.978 | 2.995 | 2.972 | 3.030 | 3.032 | 2.988 | 2.993 | 3.009 | 3.010 | 2.998 | 3.013 |
| $Ti$ | 0.000 | 0.004 | 0.000 | 0.004 | 0.004 | 0.002 | 0.000 | 0.000 | 0.008 | 0.003 | 0.001 | 0.003 | 0.002 | 0.002 | 0.000 | 0.000 | 0.001 | 0.002 | 0.001 | 0.002 |
| $Al$ | 2.028 | 1.983 | 2.023 | 1.999 | 2.051 | 2.067 | 2.082 | 2.054 | 2.053 | 2.101 | 2.070 | 2.103 | 2.042 | 2.054 | 2.036 | 2.031 | 2.036 | 2.034 | 2.038 | 2.023 |
| $Fe^{3+}$ | 0.000 | 0.037 | 0.000 | 0.000 | 0.000 | 0.000 | 0.000 | 0.000 | 0.000 | 0.000 | 0.000 | 0.000 | 0.000 | 0.000 | 0.000 | 0.000 | 0.000 | 0.000 | 0.000 | 0.000 |
| $Fe^{2+}$ | 1.910 | 1.585 | 1.959 | 1.587 | 2.231 | 2.113 | 2.196 | 2.226 | 1.797 | 1.730 | 1.867 | 1.801 | 1.984 | 2.079 | 2.068 | 2.078 | 2.250 | 2.289 | 2.143 | 2.205 |
| $Mn$ | 0.421 | 0.110 | 0.410 | 0.118 | 0.046 | 0.043 | 0.044 | 0.047 | 0.104 | 0.054 | 0.105 | 0.058 | 0.096 | 0.108 | 0.108 | 0.110 | 0.033 | 0.035 | 0.103 | 0.100 |
| $Mg$ | 0.265 | 0.161 | 0.271 | 0.155 | 0.414 | 0.278 | 0.424 | 0.379 | 0.362 | 0.384 | 0.372 | 0.364 | 0.692 | 0.549 | 0.626 | 0.576 | 0.538 | 0.486 | 0.538 | 0.468 |
| $Ca$ | 0.385 | 1.133 | 0.344 | 1.133 | 0.233 | 0.461 | 0.225 | 0.274 | 0.673 | 0.746 | 0.588 | 0.681 | 0.143 | 0.176 | 0.174 | 0.213 | 0.133 | 0.144 | 0.180 | 0.188 |
| $X_{alm}$ | 0.641 | 0.530 | 0.656 | 0.530 | 0.763 | 0.730 | 0.703 | 0.734 | 0.592 | 0.575 | 0.622 | 0.599 | 0.623 | 0.661 | 0.689 | 0.693 | 0.740 | 0.751 | 0.714 | 0.698 |
| $X_{prp}$ | 0.089 | 0.054 | 0.091 | 0.052 | 0.142 | 0.096 | 0.136 | 0.125 | 0.119 | 0.127 | 0.124 | 0.121 | 0.217 | 0.174 | 0.209 | 0.192 | 0.177 | 0.159 | 0.179 | 0.153 |
| $X_{sps}$ | 0.141 | 0.037 | 0.137 | 0.039 | 0.016 | 0.015 | 0.014 | 0.016 | 0.034 | 0.018 | 0.035 | 0.019 | 0.030 | 0.034 | 0.036 | 0.037 | 0.011 | 0.012 | 0.034 | 0.033 |
| $X_{grs}$ | 0.129 | 0.379 | 0.115 | 0.379 | 0.080 | 0.159 | 0.072 | 0.090 | 0.221 | 0.248 | 0.196 | 0.226 | 0.045 | 0.056 | 0.058 | 0.071 | 0.044 | 0.047 | 0.060 | 0.061 |
| $X_{Mg}$ | 0.122 | 0.092 | 0.122 | 0.089 | 0.156 | 0.116 | 0.162 | 0.146 | 0.168 | 0.182 | 0.166 | 0.168 | 0.259 | 0.209 | 0.232 | 0.217 | 0.193 | 0.175 | 0.201 | 0.175 |

\* EDS SEM analyses

**Table 3: Representative garnet analyses showing re-calculated garnet compositions on the basis of 8 cations and 12 oxygens.**





| Mineral | Muscovite | | | | | | Biotite | | | | | | | | | Plagioclase | | | | | | |
|---|---|---|---|---|---|---|---|---|---|---|---|---|---|---|---|---|---|---|---|---|---|---|
| Sample | UL248 | UL248 | AR71 | AR153 | AR26 | AR25b | UL248 | UL248 | AR71 | AR153 | AR153 | AR26 | AR26 | AR25b | AR25b | UL248 | UL248 | AR71 | AR153 | AR26 | AR25b | AR25b |
| | Early | Late | | | | Late | Grt in. | Mtx | | Early | Late | Grt in. | Mtx | Restite | Leuco | Core | Rim | | | | Restite | Leuco |
| **Wt%** | | | | | | | | | | | | | | | | | | | | | | |
| $SiO_2$ | 46.32 | 48.26 | 46.69 | 48.33 | 45.72 | 45.31 | 36.79 | 36.29 | 36.98 | 38.06 | 38.26 | 37.82 | 36.18 | 36.04 | 36.05 | 61.17 | 62.75 | 64.92 | 60.48 | 63.01 | 61.51 | 61.79 |
| $TiO_2$ | 1.08 | 1.00 | 0.84 | 0.94 | 1.27 | 1.79 | 3.53 | 2.95 | 2.33 | 1.93 | 0.78 | 3.58 | 4.58 | 3.84 | 3.67 | 0.00 | 0.00 | 0.00 | 0.00 | 0.00 | 0.02 | 0.01 |
| $Cr_2O_3$ | 0.03 | 0.03 | 0.02 | 0.02 | 0.04 | 0.05 | 0.01 | 0.01 | 0.05 | 0.10 | 0.03 | 0.00 | 0.00 | 0.11 | 0.02 | 0.00 | 0.00 | 0.00 | 0.00 | 0.00 | 0.00 | 0.00 |
| $Al_2O_3$ | 33.27 | 30.78 | 34.06 | 32.99 | 34.67 | 34.24 | 17.33 | 16.94 | 18.57 | 20.60 | 19.53 | 20.57 | 17.69 | 18.74 | 19.36 | 23.95 | 23.01 | 23.27 | 25.40 | 23.34 | 24.60 | 24.95 |
| $Fe_2O_3$ | 2.15 | 0.84 | 0.00 | 0.00 | 0.00 | 1.26 | 0.00 | 0.00 | 0.00 | 0.00 | 0.00 | 0.00 | 0.00 | 0.00 | 0.00 | 0.04 | 0.20 | 0.00 | 0.01 | 0.00 | 0.00 | 0.00 |
| $FeO$ | 0.47 | 1.83 | 1.40 | 1.63 | 1.06 | 0.27 | 21.55 | 21.26 | 19.44 | 15.98 | 16.50 | 13.80 | 18.48 | 17.95 | 17.52 | 0.00 | 0.00 | 0.00 | 0.00 | 0.00 | 0.00 | 0.04 |
| $MnO$ | 0.02 | 0.05 | 0.02 | 0.00 | 0.00 | 0.00 | 0.23 | 0.22 | 0.05 | 0.05 | 0.04 | 0.01 | 0.00 | 0.04 | 0.06 | 0.00 | 0.00 | 0.00 | 0.05 | 0.06 | 0.06 | 0.03 |
| $MgO$ | 1.09 | 1.76 | 1.44 | 1.61 | 1.07 | 1.16 | 6.82 | 7.81 | 11.46 | 11.69 | 13.59 | 11.62 | 9.13 | 11.45 | 10.33 | 0.00 | 0.00 | 0.01 | 0.00 | 0.03 | 0.00 | 0.00 |
| $CaO$ | 0.00 | 0.00 | 0.00 | 0.00 | 0.00 | 0.00 | 0.02 | 0.00 | 0.00 | 0.00 | 0.01 | 0.09 | 0.00 | 0.00 | 0.00 | 5.48 | 4.44 | 3.69 | 5.59 | 4.95 | 5.85 | 6.16 |
| $Na_2O$ | 0.36 | 0.25 | 1.11 | 0.76 | 0.40 | 0.34 | 0.10 | 0.07 | 0.22 | 0.19 | 0.17 | 0.00 | 0.09 | 0.16 | 0.11 | 8.58 | 9.34 | 9.18 | 8.34 | 8.61 | 8.16 | 7.93 |
| $K_2O$ | 11.25 | 11.30 | 9.46 | 10.17 | 10.55 | 11.14 | 10.03 | 10.16 | 8.17 | 8.94 | 9.03 | 9.09 | 8.89 | 10.03 | 10.18 | 0.22 | 0.22 | 0.05 | 0.07 | 0.18 | 0.28 | 0.23 |
| Total | 96.03 | 96.10 | 95.04 | 96.46 | 94.77 | 95.57 | 96.41 | 95.70 | 97.27 | 97.54 | 97.93 | 96.58 | 95.04 | 98.36 | 97.29 | 99.43 | 99.96 | 101.12 | 99.94 | 100.18 | 100.48 | 101.14 |
| | | | | | | | | | | | | | | | | | | | | | | |
| Si | 3.086 | 3.216 | 3.112 | 3.187 | 3.069 | 3.026 | 2.906 | 2.870 | 2.820 | 2.860 | 2.839 | 2.873 | 2.864 | 2.712 | 2.749 | 2.726 | 2.772 | 2.843 | 2.683 | 2.792 | 2.721 | 2.721 |
| Ti | 0.054 | 0.050 | 0.042 | 0.047 | 0.064 | 0.090 | 0.210 | 0.175 | 0.134 | 0.109 | 0.044 | 0.204 | 0.273 | 0.217 | 0.210 | 0.000 | 0.000 | 0.000 | 0.000 | 0.000 | 0.001 | 0.000 |
| Cr | 0.001 | 0.001 | 0.001 | 0.001 | 0.002 | 0.003 | 0.001 | 0.000 | 0.003 | 0.006 | 0.002 | 0.000 | 0.000 | 0.006 | 0.001 | 0.000 | 0.000 | 0.000 | 0.000 | 0.000 | 0.000 | 0.000 |
| Al | 2.613 | 2.418 | 2.676 | 2.564 | 2.743 | 2.695 | 1.614 | 1.579 | 1.669 | 1.824 | 1.708 | 1.842 | 1.651 | 1.662 | 1.740 | 1.258 | 1.198 | 1.201 | 1.328 | 1.219 | 1.283 | 1.295 |
| $Fe^{3+}$ | 0.108 | 0.042 | 0.000 | 0.000 | 0.000 | 0.063 | 0.000 | 0.000 | 0.000 | 0.000 | 0.000 | 0.000 | 0.000 | 0.000 | 0.000 | 0.001 | 0.007 | 0.000 | 0.000 | 0.000 | 0.000 | 0.001 |
| $Fe^{2+}$ | 0.026 | 0.102 | 0.078 | 0.090 | 0.059 | 0.015 | 1.424 | 1.406 | 1.240 | 1.004 | 1.024 | 0.877 | 1.223 | 1.129 | 1.117 | 0.000 | 0.000 | 0.000 | 0.000 | 0.000 | 0.000 | 0.001 |
| Mn | 0.001 | 0.003 | 0.001 | 0.000 | 0.000 | 0.000 | 0.015 | 0.015 | 0.003 | 0.003 | 0.003 | 0.001 | 0.000 | 0.003 | 0.004 | 0.000 | 0.000 | 0.000 | 0.002 | 0.002 | 0.002 | 0.001 |
| Mg | 0.108 | 0.175 | 0.143 | 0.159 | 0.107 | 0.115 | 0.803 | 0.920 | 1.303 | 1.309 | 1.503 | 1.316 | 1.077 | 1.284 | 1.174 | 0.000 | 0.000 | 0.000 | 0.000 | 0.001 | 0.000 | 0.000 |
| Ca | 0.000 | 0.000 | 0.000 | 0.000 | 0.000 | 0.000 | 0.002 | 0.000 | 0.000 | 0.000 | 0.001 | 0.007 | 0.000 | 0.000 | 0.000 | 0.262 | 0.210 | 0.173 | 0.266 | 0.235 | 0.277 | 0.291 |
| Na | 0.046 | 0.033 | 0.143 | 0.097 | 0.052 | 0.044 | 0.015 | 0.010 | 0.033 | 0.028 | 0.024 | 0.000 | 0.014 | 0.023 | 0.016 | 0.741 | 0.800 | 0.779 | 0.717 | 0.740 | 0.700 | 0.677 |
| K | 0.956 | 0.961 | 0.804 | 0.855 | 0.903 | 0.949 | 1.011 | 1.025 | 0.795 | 0.857 | 0.855 | 0.881 | 0.898 | 0.963 | 0.990 | 0.012 | 0.012 | 0.003 | 0.004 | 0.010 | 0.016 | 0.013 |
| | | | | | | | | | | | | | | | | | | | | | | |
| $X_{Mg}$ | | | | | | | 0.361 | 0.396 | 0.512 | 0.566 | 0.595 | 0.600 | 0.468 | 0.532 | 0.512 | | | | | | | |
| An % | | | | | | | | | | | | | | | | 26 | 21 | 18 | 27 | 24 | 28 | 30 |

**Table 4: Representative biotite, feldspar and white mica compositions.**



### 4.2.2 P-T Modelling

Garnet cores represent an earlier P-T history, shielded from re-equilibration with the matrix by garnet rims during subsequent metamorphism, and the conditions for garnet core and rim formation were therefore modelled separately. Garnet core formation was modelled using the composition obtained from the bulk rock XRF analysis (Table S1). Since leucosome was

observed in less sheared outcrops of the same rock type within 10 m of the sample site, garnet cores in the sample were modelled as part of a migmatite assemblage. The pressure was estimated at 10 kbar based on the position of garnet isopleths representing measured garnet core compositions in a water-saturated pseudosection. The water content was then estimated from the position of the solidus in a T-$X_{H2O}$ pseudosection calculated at 10 kbar. In Figure 7A modelled $X_{Mg}$, Grs and Sps compositions fit measured garnet core compositions within the phase field Grt-Bt-Pl-Kfs-Ms-Rt-Qtz-Melt, constraining pre-

S2 conditions to 705-735 ºC and 9.9-10.8 kbar. As diffusion in garnet at upper amphibolite facies conditions is considered to be fast (e.g. Caddick et al., 2010), care should be taken with interpreting these as absolute P-T conditions as garnet cores could have been modified by diffusion during the overprinting event. However, we interpret the shape of the garnet profile to indicate that diffusion is reflected by the transition zone between Grt1 cores and Grt2 rims, and that the flat cores probably represent Grt1 that escaped diffusion during S2 overprinting (Fig. 6A). The interpretation is consistent with the presence of rutile and

lack of titanite as inclusions in garnet cores (Fig. 4C, 6A). It also agrees well with the anorthite ($An_{25-29}$) content in zoned plagioclase cores and Si content of large Ms1 mica fish (Table 4).

The P-T estimate for formation of garnet rims and S2 (Fig. 7B) was calculated with the XRF bulk rock composition from which garnet cores were subtracted by using the modal Grt1 proportion estimated from the pseudosection (0.75 modal%). LOI

from the bulk rock analysis was used as $H_2O$ content. Modelled $X_{Mg}$, Sps and Grs compositions fit measured garnet rim compositions in the phase field Grt-Bt-Ms-Pl-Kfs-Ttn-Rt-Qtz, and constrain S2 shearing to 635-690 °C and 11.5–12.3 kbar (Fig. 7B). Anorthite content in plagioclase rims, $X_{Mg}$ content in matrix biotite, and Si content of Ms2 agree well with this estimate. Titanite is predicted as part of this assemblage, which is consistent with the large amount of titanite in the matrix and as inclusions in Grt2 (Fig. 4C, 6A). Based on field structural observations we interpret this P-T estimate for S2 to represent

shearing along the KNC-Vaddas boundary, and titanite growth is directly related to this event.

### 4.3 Upper Vaddas Nappe (AR71)

#### 4.3.1 Petrography and mineral chemistry

Sample AR71 (Fig 2, 3; Table 1) is a medium-grained garnet-mica-schist with a strong S2 foliation defined by intergrown biotite and muscovite (Fig. 4F). Muscovite is the dominant mica and occurs as large (1-3 mm long) mica fish. Plagioclase is

found as rare porphyroclasts and as single recrystallized grains within quartz layers. Rutile is abundant as elongate grains parallel to the foliation (Fig. 4F). Minor ilmenite occurs as rims on rutile grains. Garnet grains are found as small (0.03-0.1 mm), single, idiomorphic grains and as larger (1-2 mm) fish-shaped clusters of grains parallel to the foliation, with a



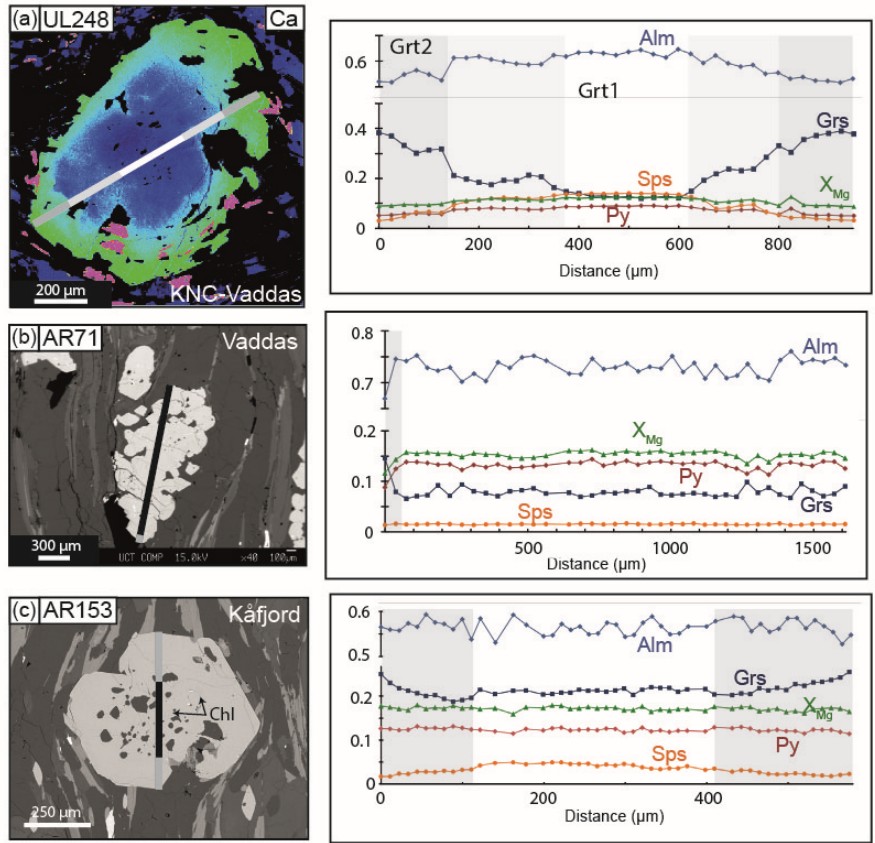

**Figure 6: BSE images, compositional maps and garnet profiles from the Vaddas and Kåfjord nappes. (a) Garnet from the KNC-Vaddas boundary, sample UL248, displays strong zoning with defined core (Grt1) and rim (Grt2) zones and a transition zone between them. Pink grains in garnet rims are titanite. (b) Garnet from the Vaddas Nappe, sample AR71, has a flat profile with**
5  **occasional thin growth rims. (c) Garnet from the Kåfjord Nappe, sample AR153, has a profile that shows some zoning, with mainly Grs and Sps contents displaying a difference between inclusion-rich cores and inclusion-poor rims.**

synkinematic geometry displaying typical Caledonian top-to-SE shear sense (Fig. 4F). Garnet shows an almost flat compositional profile with slight variations between cores, intermediate zones and rims. In rare cases fish-shaped garnets have a thin rim of a different composition. These rims only occur at the apex of fish-shaped garnets and are never present on the S2

10  parallel rims, indicating that they represent growth of garnet during late S2 shearing (e.g. Fig. 6B, Table 3). In garnets that do not display the thin rim $X_{Mg}$ varies from 0.17 in the cores to 0.14 in the intermediate zones and garnet edges. Grs is slightly lower in cores ($Grs_{7-8}$) than intermediate zones and edges ($Grs_{9-10}$). Sps content varies between $Sps_{1.4-1.6}$. Almandine content varies between $Alm_{68-73}$ (Fig. 6B, Table 3). The thin garnet rims display the following composition: $Alm_{65-67}$ $Grs_{14-16}$ $Sps_{1.2}$ and





an $X_{Mg}$ of ~0.11 (e.g. Fig. 6B, Table 3). $X_{Mg}$ in biotite ranges between 0.50 and 0.53, Si content in muscovite is between 3.05 and 3.11 (a.p.f.u.), and plagioclase has an anorthite content of $An_{18-20}$ (Table 4).

### 4.3.2 P-T Modelling

Modelled $X_{Mg}$, Grs and Sps isopleths for cores, intermediate zones and normal rims corresponding to the observed composition
of garnet and biotite intersection in the Grt-Bt-Pl-Ms-Pg-Qtz-Rt-H$_2$O phase field (Fig. 7C). This is consistent with the assemblage present in the sample except for paragonite, which was not observed but is predicted to be present only in small amounts (< 2 modal%). The slightly lower Grs, and higher $X_{Mg}$ contents of the cores indicate that minor garnet zoning resulted from growth during an increase in pressure (Fig. 7C). Si content in muscovite was used to better constrain the pressure range due to the pressure insensitive orientation of garnet isopleths in the phase field. Together with garnet compositions, the Si
content of muscovite constrains conditions of 630-640 ºC and 11.7-13 kbar for S2 metamorphism and shearing (Fig. 7C), with cores forming between 11.7 and 12.2 kbar and intermediate zones and normal rims forming at up to ~13 kbar. The higher Grs and lower $X_{Mg}$ contents, and a similar Sps content in the rare thin rims on garnet, together with the growth of ilmenite rims on rutile, indicate retrogression to lower temperature, and probably lower pressure conditions (Fig. 7C). The lack of an older foliation in the sample or in the outcrop, and the zoning pattern of garnet leads us to interpret that the Vaddas Nappe was only
affected by S2 Caledonian metamorphism and subsequently retrogressed to lower P-T conditions.

### 4.4 Kåfjord Nappe (AR153)

#### 4.4.1 Petrography and mineral chemistry

Sample AR153 (Fig. 2, 3; Table 1) is a medium- to fine-grained garnet-biotite-zoisite-schist taken from the lower Kåfjord Nappe. Biotite, muscovite, zoisite and elongate quartz-feldspar aggregates define a strong mylonitic S2 foliation (Fig. 4I).
Muscovite generally occurs within the foliation grown together with biotite and as rare grains crosscutting the S2 foliation. Garnet is porphyroblastic and idiomorphic and sometimes has inclusion-rich cores and inclusion-poor rims. Cores include quartz, biotite, ilmenite and chlorite (Fig. 4H). Chlorite is absent in the matrix. Rutile and titanite are both found as elongate grains parallel to the foliation, although rutile is significantly more abundant. Two generations of ilmenite occur as small inclusions in garnet and as thin rims on rutile grains (Fig. 4H, 4I). Garnet shows some compositional change from core to rim
(Fig. 6C; Table 3). Alm content is variable across the garnet (between $Alm_{53-62}$). Grs content is around 19-21 mol% in the cores and increases towards the rims up to 22-26 mol%. In the profile (Fig. 6C) $X_{Mg}$ does not vary much between cores and rims (0.15-0.18) although spot analyses show that $X_{Mg}$ is often slightly lower in the cores than in the rims (Table 3). The Sps content is higher in the cores (3-5 mol%) than in the garnet rims (1-2 mol%). Plagioclase shows no zoning, and has a composition of $An_{25}$-$An_{32}$. $X_{Mg}$ in biotite is between 0.56 and 0.59. Si content of the white mica is between 3.16 and 3.2
(a.p.f.u.).





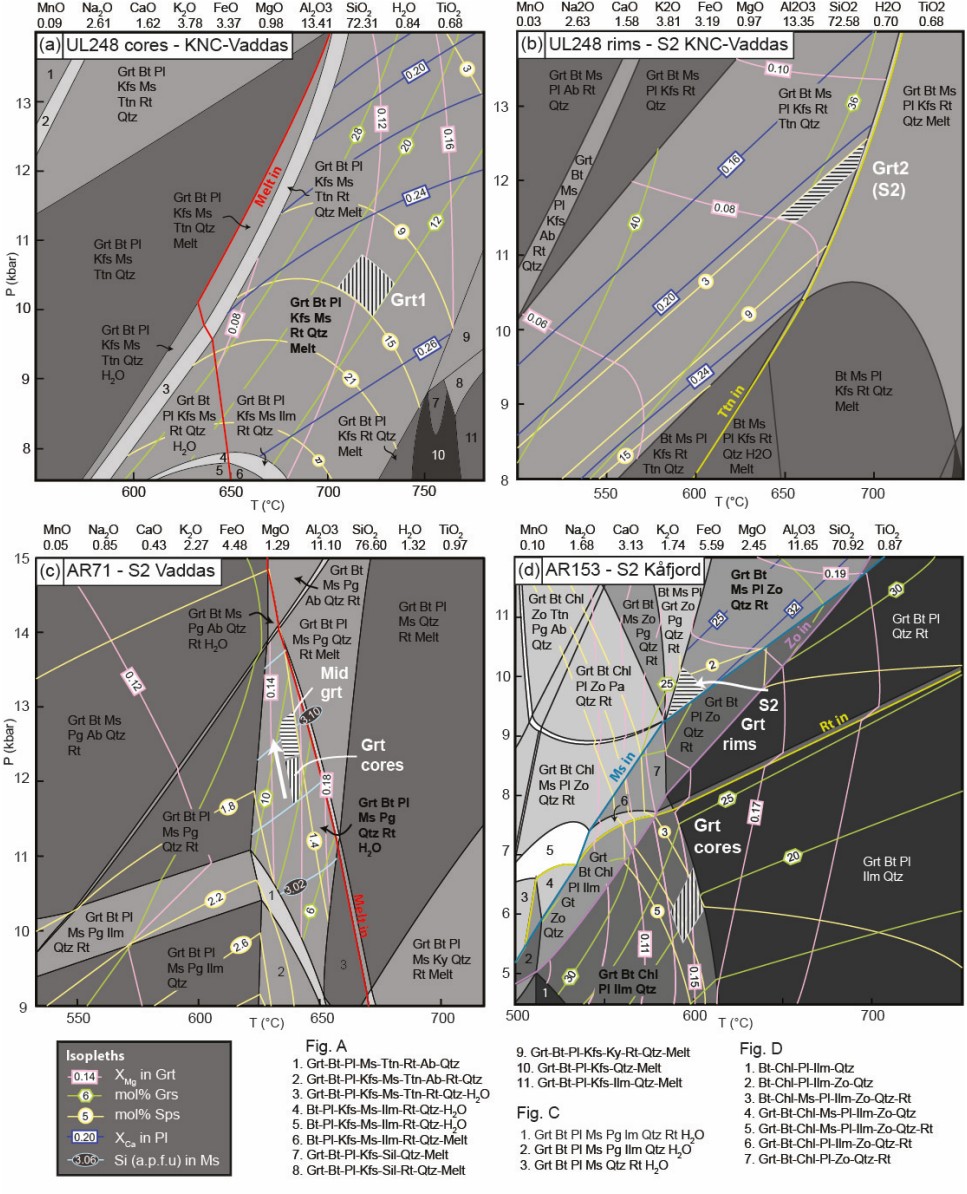

**Figure 7: Pseudosections describing metamorphism at the Kalak-Vaddas nappe boundary (sample UL248) and in the Vaddas (sample AR71) and Kåfjord (sample AR153). Estimates in the pseudosections are displayed as striped boxes. (a) Garnet core (Grt1) growth conditions were estimated from the $X_{Mg}$, Grs and Sps contents of garnet cores. Isopleths for anorthite content are shown,**





and are in agreement with measured anorthite content in plagioclase cores. The pseudosection was calculated using the bulk composition determined from XRF analysis of the whole rock. (b) S2 shearing conditions were estimated from $X_{Mg}$, Grs and Sps content correlating with measured garnet rims (Grt2). The estimate correlates well with anorthite content in plagioclase rims. The pseudosection was calculated using a bulk composition from which garnet cores were subtracted. (c) Grs, Sps, and $X_{Mg}$ contents in garnet give core and rim at slightly lower and higher pressures, respectively. Si (a.p.f.u) in muscovite was used to further constrain pressures. (d) Grs, Sps and $X_{Mg}$ contents in zoned garnets from the Kåfjord Nappe give a garnet core estimate at lower pressure conditions, and garnet rim estimate at higher pressure conditions. Chlorite and ilmenite inclusions in garnet cores are consistent with the garnet core estimate.

### 4.4.2 P-T Modelling

Zoning in garnet and the difference between garnet inclusion assemblage and matrix assemblage reflect growth at evolving P-T conditions. The presence of chlorite and ilmenite as inclusions in garnet (Fig. 4H, I) indicate that the cores grew at different conditions relative to rims and the matrix, limiting core growth below 620 ºC and 8.5 kbar. Garnet cores show a relatively higher Sps content and lower content of Grs than rims while $X_{Mg}$ shows little variation between cores and rims. The sample was modelled under water saturated conditions and the resulting pseudosection is shown in Figure 7D. The presence of chlorite and ilmenite as inclusions in garnet cores limits core growth conditions below 620 ºC and 8.5 kbar. Modelled $X_{Mg}$, Grs and Sps isopleths are consistent with measured garnet core composition within the phase field Grt-Bt-Chl-Pl-Ilm-Qtz, and constrains garnet core growth to 590 – 610 °C and 5.5-6.8 kbar (Fig. 7D). The presence of rutile, zoisite and muscovite in the matrix constrains S2 conditions above 9 kbar in the phase field Grt-Bt-Ms-Pl-Zo-Rt, consistent with the matrix assemblage (Fig. 4I). Modelled $X_{Mg}$, Grs and Sps isopleths representing measured garnet rim compositions constrain P-T conditions to 580-605 ºC and 9.2-10.1 kbar. The morphology of the isopleths is consistent with a lack of significant zoning in pyrope (and $X_{Mg}$ content) in the garnets. A match of model compositions for the rim estimate is also observed with the measured plagioclase composition (Fig. 7D) and measured Si content in muscovite. The $X_{Mg}$ value of biotite in the sample has a large range that is consistent with modelled values for both the garnet core and rim estimates, and which is likely the result of biotite growth and equilibration over evolving P-T conditions. There is no microstructural evidence to suggest that there is more than one generation of biotite. The shape of the garnet compositional profile (Fig. 6C) compared to the morphology of the isopleths on the pseudosection suggests initial garnet growth at low pressures followed by continuous garnet growth with increasing pressure (and slightly increasing temperature). Ilmenite rims on rutile grains are likely a result of later retrogression to lower pressures and temperatures.

### 4.5 Nordmannvik Nappe – migmatite (AR25b)

### 4.5.1 Petrography and mineral chemistry

Sample AR25b (Figs. 2, 3; Table 1) is a coarse-grained migmatitic paragneiss with leucosome segregations and dark restitic layers (Figs. 5E-F). It has a macroscopic weak foliation defined by biotite and migmatitic banding. The leucosome consists of plagioclase, quartz and K-feldspar (partly replaced by myrmekite). Kyanite and garnet porphyroblasts (2-3 mm in size) are common along leucosome boundaries and in the restite (Fig. 5F). Kyanite occurs as large (up to 6 mm) porphyroblasts oriented



both randomly and parallel to the foliation and almost always in association with biotite, and often with quartz (Fig. 5F). Kyanite crystals sometimes contain inclusions of quartz and biotite, and are occasionally in contact with garnet. Smaller kyanite crystals are also found as inclusions in quartz in leucosomes. Garnet is usually idiomorphic, and contains inclusions of biotite, quartz, ilmenite and sillimanite (the latter confirmed by Raman spectra; Figs. 8, 9A). In minor amounts, fine-grained sillimanite

5   is intergrown with biotite or occurs along kyanite and garnet rims. Muscovite occurs as rare small grains overgrowing biotite and is often associated with fine-grained sillimanite. Garnets show a relatively flat compositional profile in the core and gradual changes near the rims, probably indicative of diffusion zoning. $X_{Mg}$ in the cores is 0.23-0.26 and decreases to 0.20-0.23 in the rims. Alm in cores is between 65-69 mol% and 69-71 mol% in the rims. Grs in garnet cores is as low as 4.5 mol% and at the rims 6-7 mol%. Sps content of garnet cores is 3 mol% and 3.5 - 4 mol% in the rims (Fig. 9A).

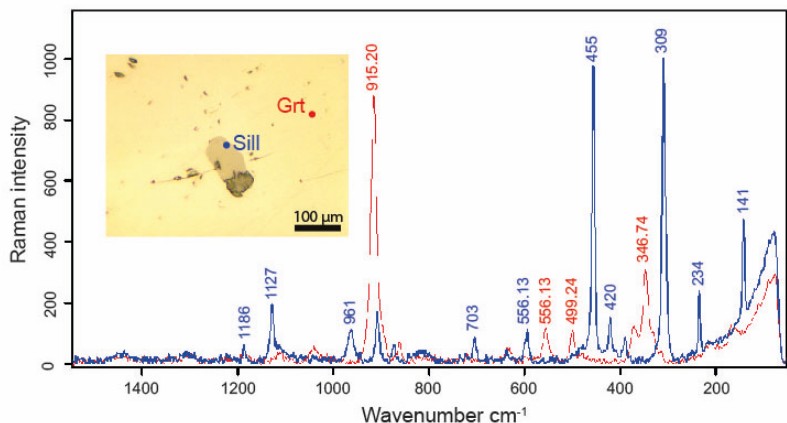

**Figure 8: Raman spectra for a sillimanite inclusion from a garnet core in the Nordmannvik migmatite (sample AR25b).**

**4.5.2 P-T Modelling**

Metamorphic conditions for the migmatitic S1 foliation in the Nordmannvik Nappe were estimated from sample AR25b (Fig. 10A). To set water content for migmatisation first an approximate pressure for equilibration was estimated from a water-

15   saturated pseudosection using the XRF bulk composition. Isopleths for Grs, Sps and $X_{Mg}$ contents intersect at ~10 kbar. Water content used in the final pseudosection (Fig. 10A) was then determined from the position of the solidus on a T-$X_{H2O}$ pseudosection calculated at 10 kbar for the mineral assemblage associated with melting. The presence of kyanite and sillimanite, together with the variation in plagioclase and biotite composition, suggest that the minerals record evolving metamorphic conditions. The flat profile for garnet cores and changes at the rims suggests that the rims may have been modified

20   by diffusion due to disequilibrium (Fig. 9A). The association of kyanite, biotite and garnet with the leucosome in the S1 foliation suggests that these minerals formed the stable assemblage during S1 partial melting. Given the large volume of kyanite associated with the leucosome it can be explained by the following reaction: Ms ± Pl + Qtz = Kfs + Als + Liq. This is



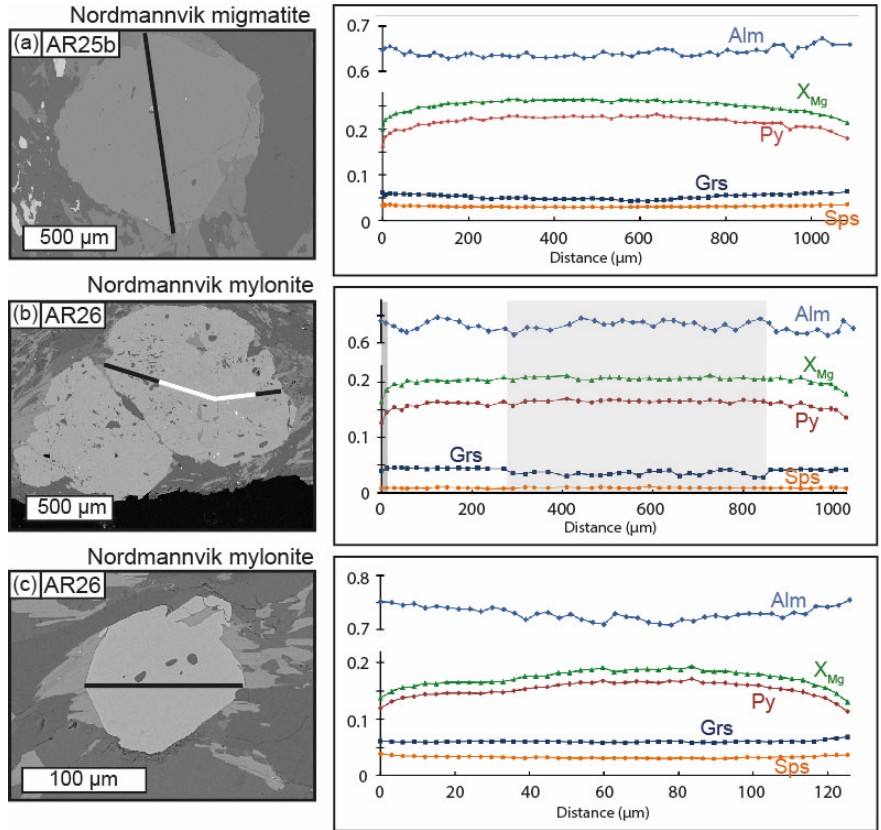

**Figure 9: BSE images and garnet profiles for Nordmannvik Nappe samples AR25b and AR26. (a) Garnet in AR25b displays a profile with a steady increase in Alm, Grs and Sps contents and decrease in Py and X$_{Mg}$ towards garnet rims. (b) The garnet profile for the large garnet in sample AR26 displays a relatively flat profile with some compositional change towards the rims. An inclusion-free core, marked in grey on the profile and in white on the BSE image, shows slightly lower and more variable Grs content. (c) The garnet profile for small garnets in sample AR26 shows a steady increase in Alm, Grs and Sps contents and steady decrease in Py and X$_{Mg}$ contents towards the rims.**

consistent with the lack of muscovite in the rock and presence of K-feldspar. Based on the relationship between garnet rims

and the leucosome, garnet rim compositions were taken as representative of the S1 migmatization conditions.

Most measured garnet rim compositions and some intermediate zone compositions correspond with modelled isopleths in the phase field Grt-Bt-Pl-Kfs-Ky-Qtz-Melt on the pseudosection (Fig. 10A). X$_{Mg}$, Grs and Sps contents from these analyses constrain S1 partial melting to 760 – 790 °C and 9.4 – 11 kbar. Several measured garnet rims also correspond with modelled isopleths below the solidus within the muscovite-bearing assemblage, consistent with cooling, late muscovite growth and late



minor diffusional resetting of garnet rims. The minor fine-grained sillimanite with biotite and along kyanite and garnet rims is consistent with retrogression to lower pressures. The homogenous garnet core compositions were also plotted on the pseudosection (Fig. 10A) and modelled isopleths for Grs, Sps and $X_{Mg}$ contents correspond with measured contents in the sillimanite-bearing phase field (Grt-Bt-Pl-Kfs-Sill-Qtz-Melt) above the solidus at 790 – 815 °C and 8.9 – 9.9 kbar (Fig. 10A).

5  These conditions correspond to the temperatures during S1 melting, but at slightly lower pressures. The range in measured $X_{Mg}$ content of biotite corresponds with isopleths that plot consistently with both estimates, whereas the measured anorthite content for matrix plagioclase corresponds with isopleths that plot toward the lower pressure conditions (sillimanite-bearing assemblage). It should be noted that the conditions estimated for garnet core formation may be affected by re-equilibration of the cores by later diffusion.

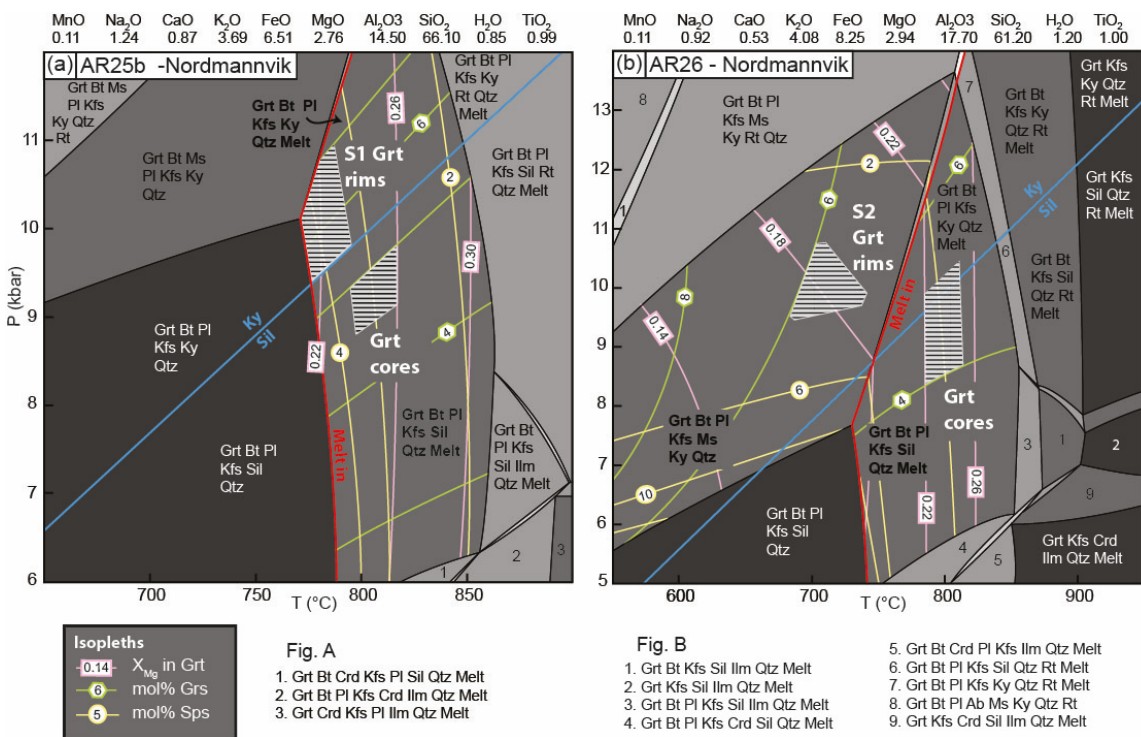

**Figure 10: Pseudosections for the Nordmannvik Nappe. (a) Measured garnet core and rim compositions for migmatite (AR25b) agree with modelled compositions that plot above the solidus. Garnet cores give an apparent estimate in the sillimanite stability field. Garnet rims record S1-related partial melting in the kyanite stability field. (b) Measured compositions of garnet cores and rims for S2 mylonite (AR26) are shown. Garnet cores give an apparent estimate above the solidus in the sillimanite stability field (S1 melting). Garnet rims and small garnets give an estimate below the solidus for S2-shearing.**





### 4.6 Nordmannvik Nappe – mylonite (AR26)

#### 4.6.1 Petrography and mineral chemistry

Sample AR26 (Figs. 2, 3; Table 1) is a fine-grained garnet-kyanite micaschist that displays a pervasive mylonitic S2 foliation defined by biotite, muscovite, and kyanite (Fig. 5B). It has a strong L2 stretching lineation defined by quartz aggregates.

Plagioclase locally forms 0.1-0.3 mm-sized sigma clasts. Biotite occurs as two generations; less common large foliation-parallel grains, and as finer-grained elongate grains intergrown with muscovite. Muscovite occurs as mica fish parallel to the foliation, and as rare late grains that crosscut the foliation (Fig. 5B). Garnet is found as large 1-2 mm idiomorphic porphyroblasts (Fig. 9B) that sometimes have inclusion-poor cores (with occasional sillimanite) and more inclusion rich rims (quartz and biotite inclusions), and as small 0.1-0.4 mm idiomorphic grains that occur singularly (Fig. 9C) or as clusters

oriented parallel to the S2 foliation. The bimodal garnet grain size is also observed in hand specimen and is therefore not an apparent artifact of sectioning.

The larger garnets display a slight difference in composition relative to the smaller ones. Their $X_{Mg}$ contents are similar, however the larger garnets have slightly lower Sps and Grs contents ($Sps_{1-1.5}$, $Grs_{4-5}$) than the smaller garnets ($Sps_{2.5-3.5}$, $Grs_{5-}$

$_6$). Almandine content in the larger garnets ($Alm_{71-76}$) is higher than in the smaller garnets ($Alm_{65-72}$; Fig. 9B, C, Table 3). Thin rims of larger garnets sometimes show similar compositions to the small garnets. The garnet profiles for the large and small garnets are similar, although the cores of larger garnets have a wider flat profile. Grs is the only end-member that reflects the apparent zoning displayed by the microstructure of the larger garnets (Fig. 9B), being slightly lower in the nearly inclusion-free cores (Fig. 9B, grey box) then higher along the flat profile of the inclusion-rich rims. The larger garnets also have a thin

10-20 μm-wide rim (Fig. 9B; dark grey box) with lower $X_{Mg}$ values and Grs contents than cores. Smaller garnets have a similar profile but lack the wide flat cores, and have compositions similar to those of the rims of larger garnets (Fig. 9C). Based on the similarity in rim compositions, and the lack of a chemical zoning reflecting the microstructure, it appears that the slight compositional variations in both the large and small garnets probably resulted from diffusion during high-grade metamorphism. The smaller garnets are probably more completely re-equilibrated than the larger ones. The two generations of biotite have

different compositions. Older biotite has an $X_{Mg}$ between 0.56-0.58, closer to the composition of inclusions in garnet ($X_{Mg}$ ~0.6). Younger biotite displays $X_{Mg}$ values between 0.46-0.48. Plagioclase is unzoned with a composition of $An_{23-24}$ (Table 4).

#### 4.6.2 P-T Modelling

The variations in mineral compositions likely reflect changing P-T conditions with time. The profile shape and compositions of the larger and smaller garnets suggest that both probably started with the same homogenous composition and smaller garnets

re-equilibrated more completely than large ones by diffusion during metamorphism and S2 shearing (e.g. Caddick et al., 2010). Therefore, the cores of larger garnets may reflect the original P-T conditions. Biotite, kyanite and muscovite are in contact with garnet rims and define a clear S2 foliation associated with subsolidus shearing. $X_{Mg}$, Sps and Grs compositions of the





smaller re-equilibrated garnets and some larger garnet rims agree with modelled isopleths in the Grt-Bt-Pl-Kfs-Ms-Ky-Qtz phase field, and constrain S2 shearing to 680 – 730 °C and 9.5 – 10.9 kbar (Fig. 10B). Measured $X_{Mg}$ values for the younger biotite generation and Si content in muscovite agree well with their respective modelled isopleths for this estimate. The Grs, Sps and $X_{Mg}$ contents of larger garnet cores (e.g. Fig. 9B, light grey box) are consistent with modelled isopleths that plot within

the Grt-Bt-Pl-Kfs-Ky-Qtz-Melt and Grt-Bt-Pl-Kfs-Sil-Qtz-Melt phase fields at 790 – 810 °C and 8 – 10.4 kbar (Fig. 10B). Although garnet core compositions have been affected by later diffusion and the rock has been overprinted strongly by the S2 foliation, the estimated core conditions are consistent with those recorded by the migmatite sample (AR25b; Fig. 10A). This indicates that garnet cores record the same pre-S2 history (S1 melting and/or earlier?) reflected in the migmatites. The garnet core estimates agree with the presence of sillimanite inclusions in the cores, the measured $X_{Mg}$ content of biotite inclusions in

garnet, and anorthite content of plagioclase sigma clasts in the matrix, suggesting that garnet is not the only mineral that records this earlier P-T evolution.

## 5 Geochronology

In order to temporally constrain the metamorphic and deformation history of the RNC, zircon and titanite from several tectonic levels were selected for geochronology. The methods are described in supplement S1 and data are compiled in supplementary

tables 1-3.

### 5.1 S1 melting in the Nordmannvik Nappe – U-Pb SIMS dating of zircon

Sample AR23a (Fig. 2, 3; Table 1) was taken from an S1-lens in the Nordmannvik Nappe with clear leucosome segregations associated with kyanite-bearing restite.  The sample was cut in an attempt to obtain different zircon populations for the restite and leucosome layers. Zircons from the restite are clear, euhedral, and prismatic crystals 100- 350 μm long. All grains have

low CL-emission (dark grains) and show weak oscillatory zoning (Fig. 11A). Some grains have brighter rounded cores with discrete boundaries, indicative of a possible xenocrystic origin. Zircon from the leucosome is slightly larger than the grains from the restite. The grains are clear or yellow-orange in colour, and form euhedral prismatic crystals 150-350 μm-long. Under CL they are equally dark and show similar weak oscillatory zoning to the zircon from the restite, but inherited cores are absent (Fig. 11D). For the restite, 37 spots were analyzed in 26 grains. Five of the analyses are from the bright cores.  These cores

have Th/U ratios between 0.3-0.53. Two of the core analyses plot on concordia, two plot close, and one is strongly discordant (Fig. 11B, grey ellipses). The discordant analysis is likely due to mixing of a CL-bright core and darker rim. Of the four analyses that plot on or near concordia, the oldest three give $^{207}Pb/^{206}Pb$ dates of 1836 ± 10 Ma, 1721 ± 10 Ma, and 1599 ± 14 Ma. The fourth core gives a younger $^{206}Pb/^{238}U$ date of 585 ± 8 Ma. The remaining 32 analyses are from regions in zircon grains that show low CL-emission and weak oscillatory zoning. They have Th/U ratios between 0.09 and 0.16, overlap on

concordia and give a combined concordia age of 442 ± 2 Ma (Fig. 11C). For the leucosome, 27 spots were analyzed in 18 grains from both light and dark CL-zones in the zircon crystals. They have Th/U ratios of 0.08 to 0.19, and form a cluster on





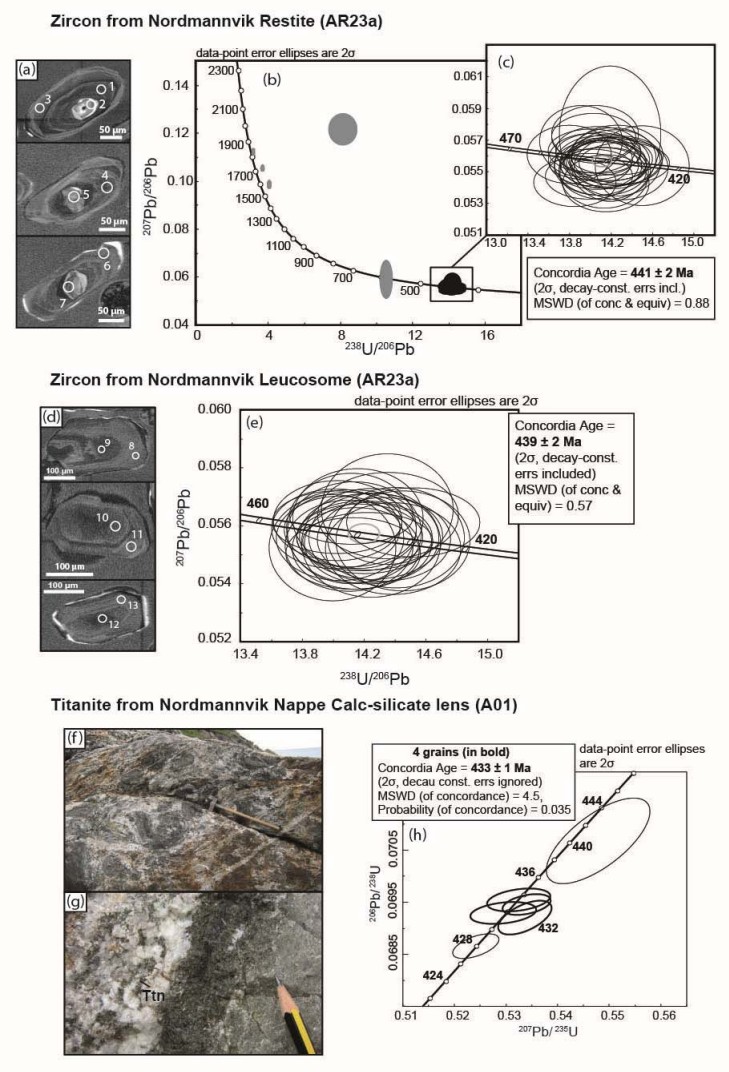

**Figure 11: Age of partial melting and metamorphism in the Nordmannvik Nappe. (a) CL images of zircon grains with single spot analyses. $^{206}$Pb/$^{238}$U dates for spot analyses are as follows: 1) 440 ± 5 Ma, 2) 585 ± 8 Ma, 3) 439 ± 5 Ma, 4) 441 ± 5 Ma, 5) 1421 ± 16 Ma, 6) 443 ± 5 Ma, 7) 1547 ± 17 Ma. Note the presence of small, inherited cores. (b) Terra-Wasserburg diagram for SIMS zircon analyses from the restite of sample AR25b. Inherited core analyses are shown as grey ellipses. (c) Concordia diagram with age calculated for the young cluster of zircons in Fig. B. (d) CL images of zircon grains from the sample with single spot analyses. $^{206}$Pb/$^{238}$U dates for spot analyses are as follows: 8) 441 ± 5 Ma, 9) 440 ± 5 Ma, 10) 441 ± 5 Ma, 11) 435 ± 5 Ma, 12) 438 ± 5 Ma, 13) 434 ± 6 Ma. No inherited cores were observed in this sample. (e) Terra-Wasserburg diagram and Concordia age for SIMS zircon analyses from the leucosome of sample AR25b. (f) Calc-silicate and leucosome mixing zone at sample site A01. (g) Close-up of calc-silicate and leucosome boundary showing titanite in a schlieren. (h) Concordia diagram for U-Pb TIMS analyses of titanites from a calc-silicate lens in the Nordmannvik Nappe (sample A01) with Concordia age calculated for 4 overlapping analyses (in bold).**



concordia giving a concordia age of 439 ± 2 Ma (Fig. 11E). The concordia age for the leucosome is slightly younger than, although within error of, the restite concordia age. The oscillatory zoning patterns in the zircons from both the restite and leucosome populations and the shape of the zircons suggest they have crystallized from melt (e.g. Hoskin and Schaltegger, 2003).

### 5.2 Metamorphism in the Nordmannvik Nappe – U-Pb dating of titanite

Sample A01 is a coarse-grained felsic rock from a low-strain lens in the Nordmannvik Nappe on Arnøya (Figs. 2, 3, 11F, G; Table 1). The sample represents an interaction zone between melt and calc-silicate rocks and is composed of K-feldspar, quartz and plagioclase and randomly oriented mafic schlieren including mainly clinopyroxene with amphibole, plagioclase, biotite, quartz, titanite, and minor calcite, ilmenite, magnetite, and iron sulphides. Only a weak macroscopic S2 foliation is present (Fig. 11F) Titanite is abundant and coarse-grained (up to 2 mm in size: Fig. 11G). It occurs as large euhedral grains mainly in the boundary zone between the schlieren and the felsic zones and is either associated with magnetite and quartz, or as inclusions in the rims of amphibole grains. Sometimes it also occurs as thick rims on ilmenite. Its microstructural relationships indicate it formed from interaction between melt and the calc-silicate rocks. Six clear-brown 300 – 600 μm sub- to euhedral titanite fragments were analyzed using the TIMS method. All analyses plot on concordia and 4 of them overlap (in bold; Fig. 11H) giving a combined concordia age of 432 ± 1 Ma. An older grain with a $^{206}Pb/^{238}U$ age of 440 ± 4 Ma and a younger grain with a $^{206}Pb/^{238}U$ age 428 ± 1 Ma do not overlap with these analyses and these were not involved in the calculation of the concordia age. Together these ages indicate a range between 444 and 427 Ma, with a peak at ~432 Ma.

### 5.3 Gabbro intrusion in the Vaddas Nappe – U-Pb SIMS dating of zircon

Sample SK18b was taken from a late-stage pegmatitic gabbro-lens within the main, medium-grained Kågen gabbro in the Vaddas Nappe (Figs. 2, 12A). It is coarse-grained and consists predominantly of up to 1 cm-long amphibole crystals in a plagioclase matrix. Zircons from the sample are euhedral, short-prismatic, and between 100-300 μm long. CL zonation is variable. Twenty spots were analysed in 16 grains. Th/U values are between 0.3 and 0.7, consistent with zircon crystallization from a melt. All analyses are concordant giving a concordia age of 439 ± 1 Ma (Fig. 12C), interpreted as the intrusive age for late-stage gabbroic pegmatites within the Kågen gabbro.

### 5.4 S2 shearing at the Vaddas-Kalak boundary – U-Pb titanite ages

Microstructural observations and P-T modeling show that titanite crystallization in sample UL248 is associated with S2 shearing and metamorphism during garnet rim growth (Fig. 4C, 6A). Titanite from the sample was therefore dated using TIMS. The majority of picked titanite grains have inclusions and the three cleanest fragments were chosen for analysis. They are pale brown 240-300 μm-long, inclusion-free grains. All three analyses plot on Concordia with $^{238}U/^{206}Pb$ dates between 436 ± 4 Ma and 431 ± 2 Ma (Fig. 12D). The mean $^{238}U/^{206}Pb$ age is calculated as 432 ± 6 Ma. We interpret this age to represent the

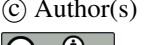



age of S2 shearing and metamorphism along the KNC-Vaddas boundary, and the likely emplacement age of the Vaddas Nappe over the KNC.

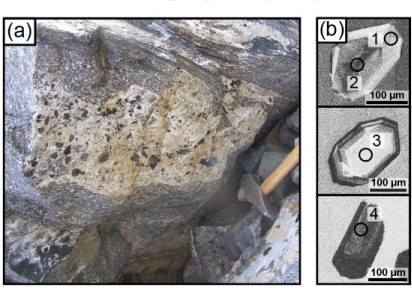

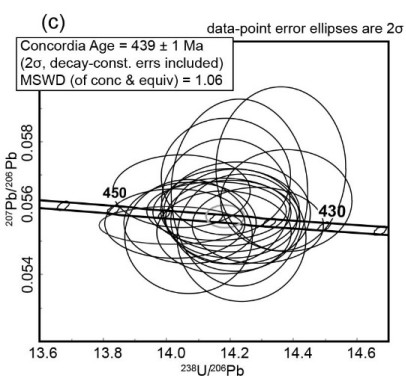

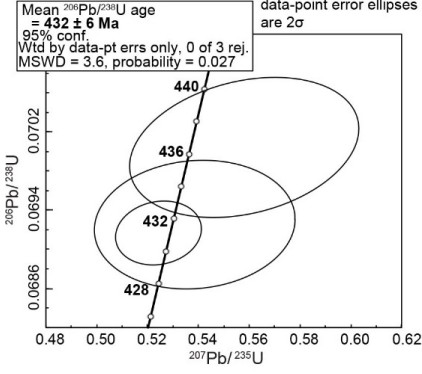

**Figure 12: Ages related to intrusion and shearing in the Vaddas Nappe. (a) Photograph of the sampled gabbroic pegmatite (sample Sk18b). Hammer head is 15 cm-long. (b) CL images of single zircon grains with spot analyses shown. The spot analyses have the following $^{206}$Pb/$^{238}$U dates: 1) 438 ± 2.5 Ma, 2) 438.2 ± 2.7 Ma, 3) 439.1 ± 2.5 Ma, and 4) 444.4 ± 2.8 Ma. (c) Terra-Wasserburg**



**diagram for SIMS zircon analysis from gabbroic pegmatite (sample SK18b) with Concordia age calculated showing the age of the Kågen gabbro intrusion into the Vaddas Nappe. (d) Concordia diagram for U-Pb TIMS analyses of titanites from the Vaddas-Kalak boundary (sample UL248) with mean $^{206}$Pb/$^{238}$U age calculated.**

**6 Interpretation: Metamorphic and magmatic evolution of the RNC**

**6.1 Potential Neoproterozoic metamorphic event in the KNC**

Clear evidence for an early metamorphic event is preserved in sample UL248 from the KNC below the Vaddas-KNC boundary. Grt1 (cores) formed at 705-735 ºC and 9.9-10.8 kbar above the solidus (Fig. 7A) at higher temperature and lower pressure than Grt2 (rims) (Fig. 7B, 13A). Sheared and dismembered leucosome suggests that melt solidified and was subsequently sheared in the solid state during the S2 overprint, indicating an early melting event being overprinted by a later sub-solidus shearing

event. Age constraints for this early melting event are lacking. However, the metamorphic conditions for Grt1 are similar to those recorded for partial melting in the KNC at Eide nearby (~730–775 °C and ~6.3–9.8 kbar; Gasser et al., 2015; Fig. 13A, B). The Eide melting event is dated at 702 ± Ma, indicating that it is of pre-Caledonian age (Gasser et al., 2015), and Caledonian migmatitization has not been recorded in the KNC so far. We therefore suggest that Grt1 in sample UL248 records the same pre-Caledonian partial melting recognized at Eide (Fig. 13B).

**6.2 Migmatization in the Nordmannvik Nappe – S1**

In the Nordmannvik Nappe early high grade metamorphism is preserved as a S1 migmatitic foliation in lower strain lenses. Phase equilibrium modelling of this foliation reveals that garnet cores in samples AR25b and AR26 reflect high temperature mid- to low- pressure partial melting in the sillimanite and kyanite stability fields. The relationship between garnet cores and rims in migmatite sample AR25b and what the conditions estimated using the garnet core compositions might represent is

difficult to interpret. Although the rim compositions are consistent with kyanite-present melting (760 – 790 °C and 9.4 – 11 kbar), the cores indicate earlier lower pressure, higher temperature conditions (790 – 815 °C and 8.9 – 9.9 kbar; Figs. 10A, 13A). The lack of two distinct generations of garnet may suggest their compositions record just one event, with their cores and rims reflecting only the most recent diffusion. If this were the case, then the estimates constrained from core and rim compositions could be interpreted to record a single migmatization event, beginning at low pressures and continuing to higher

pressures. However, such a scenario is difficult to reconcile because the melting reaction is almost completely temperature-dependent, and therefore the bulk of melting would have occurred at the lower pressure sillimanite-present conditions with no or little melt produced at higher pressure. This is in contrast to the microstructures that suggest that partial melting produced mainly kyanite. Since the garnet core compositions are likely unreliable due to high rates of diffusion that prevail at the temperatures under consideration during the S1 partial melting and S2 shearing (e.g. Caddick et al., 2010), we consider that

the bulk of the partial melting occurred in the kyanite stability field and that other possibilities for the metamorphism related to the sillimanite inclusions could be: 1) subsolidus prograde metamorphism related to S1 migmatization, or 2) some unknown pre-S1 event either above or below the solidus. Garnet cores in sample AR26 also record conditions above the solidus in the



kyanite and sillimanite stability fields (790 – 810 °C and 8 – 10.4 kbar), prior to the S2 shearing recorded by garnet rims (Figs. 10B, 13A). Polymetamorphism in the Nordmannvik Nappe has been previously related to Precambrian or Ordovician events (Elvevold et al., 1987; Lindstrøm and Andresen, 1992).

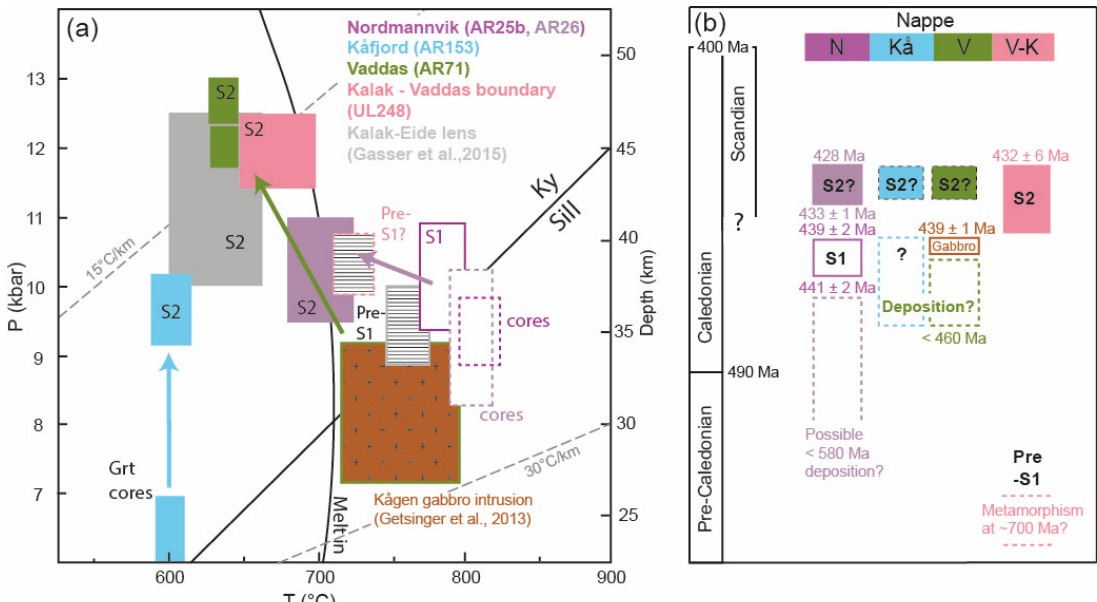

**Figure 13: Summary of P-T conditions and timing for phase equilibrium modelling and geochronological results. (a) P-T diagram comparing conditions of metamorphism for Pre-S1 garnet cores (striped boxes), S1 migmatisation (solid line box), apparent garnet core conditions in the Nordmannvik Nappe (dashed boxes), and S2 shearing (filled boxes) for all nappes. Estimated conditions for metamorphism in the KNC are shown in grey (Gasser et al., 2013). The intrusion conditions for the Kågen gabbro (brown box) are**
10 **from Getsinger et al., (2013). Arrows show anticlockwise P-T paths for the Nordmannvik Nappe rocks and Vaddas/lower Kåfjord metasediments. (b) Timing of magmatism and metamorphism comparing ages across the nappes of the RNC and placing them within a pre-Caledonian and Caledonian context. Titanite ages are shown as filled boxes and zircon ages as solid line boxes. Inferred and speculative ages are shown as dashed boxes.**

Leucosome associated with partial melting and formation of the kyanite-bearing S1 foliation crystallized between ~443 and 437 Ma (Fig. 11A-E, Fig. 13B). This indicates early Silurian melting at ~30-37 km depth with a geothermal gradient of 22-25 °C/km. The presence of leucosome veins in the axial planes of Caledonian (F2) folds in the Nordmannvik Nappe (Fig. 5D) indicates that the rocks were partially molten during folding. The rocks are also overprinted by the solid-state S2 shear foliation (Figs. 5A, C). Considered together, these structures indicate that the Nordmannvik Nappe migmatites underwent initial folding

between ~443 and 437 Ma while still partially molten, and continued shearing as the rocks solidified until the final S2 overprinting in their solid-state. This progression represents an anticlockwise P-T path (T decreasing, P increasing) from S1 to S2 conditions (Fig. 13A). There is a large range in titanite dates (444-427 Ma) for the Nordmannvik migmatite sample A01,




covering the age range of leucosome formation derived from zircon (443-437 Ma), but also extending to much younger dates. This range could be the result of either 1) diffusional Pb loss during cooling above the closure temperature of titanite (e.g. Tucker et al., 2004), or 2) protracted titanite growth (Kohn, 2017). The closure temperature of titanite is debated (e.g. Cherniak, 1993; Frost et al., 2001), although recent well-constrained work in the Western Gneiss Region suggests Pb diffusion in titanite

is slow (Spencer et al., 2013; Kohn et al., 2015) and probably ineffective below 800 °C (Kohn, 2017). The large grain size (> 300 μm) of the titanites in sample A01, and estimates for both S1 and S2 P-T conditions < 800 °C suggest it is unlikely that diffusional Pb loss caused the large age range, and from this we conclude that titanite probably grew over a protracted time period recording long-lived metamorphic processes.

The association of titanite and amphibole or titanite, magnetite and quartz suggest that it formed by rehydration (1: Cpx + Ilm + Qtz + $H_2O$ = Amph + Ttn) and/or oxidation (2: Cpx + Ilm + $O_2$ = Ttn + Mag + Qtz) reactions (e.g. Kohn, 2017) during crystallization of melt within calc-silicate layers. Both reactions are consistent with the observation of titanite rims on ilmenite grains. If both reactions were active, then prolonged titanite growth from ~444 to 427 Ma in the Nordmannvik rocks probably resulted from switching between one reaction to the other during evolution of the system, controlled by $H_2O$ and $O_2$ availability

from injected melt and metamorphic fluids. Due to the chaotic migmatite structures the rocks do not demonstrate a clear S2 shear foliation, but they record long-lived titanite growth probably driven by melt and/or fluid infiltration during prolonged early and peak Caledonian metamorphism. Both the zircon and titanite ages are consistent with the 439 ± 1 Ma metamorphic age (from zircon overgrowths) reported from the Nordmannvik Nappe at Heia (Augland et al., 2014).

### 6.3 Gabbro intrusion in the Vaddas Nappe

Coeval with the S1 migmatization in the Nordmannvik Nappe, late-stage gabbroic pegmatites intruded the Vaddas Nappe on Kågen at 439 ± 1 Ma (Fig. 12C) at pressures of 7 – 9 kbar (Getsinger et al., 2013), corresponding to a depth of 26-34 km (Fig. 13). This indicates a geothermal gradient around ~25 - 30 °C/km. The gabbroic composition clearly indicates a mantle source of the melt, and the tholeiitic composition, established from other mafic rocks and gabbros in the nappe (Lindahl et al., 2005), suggests melting in an extensional setting. The similar small gabbro bodies in the Kåfjord Nappe are probably also related to

this event. The intrusion of the Kågen gabbro at 439 ± 1 Ma is slightly earlier than intrusion of the Heia gabbro in the Nordmannvik Nappe (435 ± 1 Ma ; Augland et al., 2014).

### 6.4 Early garnet growth in the Kåfjord Nappe

Garnet cores in the sample from the lower Kåfjord Nappe (AR153) preserve lower amphibolite facies conditions (590-610 °C and 5.5 – 6.8 kbar; Figs. 7D, 13A). The P-T estimate is consistent with a slightly elevated geothermal gradient of 24 – 29

°C/km. Without direct age dating it is difficult to give the exact timing of this event. The lack of pre-S2 structures in the lower Kåfjord metasediments and likely late Ordovician depositional age constrains the time window for this metamorphism to either immediately prior to S2 shearing or during early S2 shearing (Fig. 13B). The P/T indicates a similarity to the geothermal




gradients estimated for S1 migmatization and gabbro intrusion, suggesting that initial garnet growth in the Kåfjord Nappe may record the early Silurian heating event as in the Vaddas and Nordmannvik Nappes, but at a shallower depth. Continuous garnet growth records the increase in pressure during S2 shearing and eventually peak S2 metamorphism.

### 6.5 Pervasive S2 shearing

Pervasive, amphibolite-facies S2 shearing with top-to-SE kinematics is recorded throughout the entire RNC. KNC-Vaddas boundary rocks record S2 shearing at 635-690 °C and 11.5 - 12.3 kbar at similar conditions as S2 shearing in the Vaddas PNappe (630-640 ºC and 11.7-13 kbar; Figs. 7C, 13A), at depths around ~43-46 km with the P/T ratio giving an estimated geothermal gradient of 14-15 °C/km. Garnets in the Kåfjord Nappe show an increase in pressure from 5.5 to 10.1 kbar at temperatures of 580-610°C, reflecting a change in geothermal gradient from early high temperature-low pressure conditions

to high pressure-low temperature conditions during S2 shearing (Figs. 7D, 13A). In the Nordmannvik Nappe S2 shearing is recorded in the mylonitic gneisses near the Kåfjord-Nordmannvik boundary at 680 – 730 °C and 9.5 – 10.9 kbar (Figs. 10B, 13A). Considering maximum uncertainties of ± 50 °C and ±1 kbar, the P-T estimates for S2 in the three nappes are similar, but with a trend towards the highest pressure at the base of the RNC, and the coldest temperature in the Kåfjord Nappe in the middle of the nappe stack (Fig. 13A). The age of S2 shearing along the KNC-Vaddas boundary is constrained by syn-S2 titanite

to 438-427 Ma (Figs. 12D, 13B), recording stacking of the Vaddas Nappe over the KNC. The age of S2 metamorphism is consistent with ages for Caledonian metamorphism in the equivalent Magerøy Nappe (Andersen et al., 1982; Corfu et al., 2006, 2011; Kirkland et al., 2005, 2016), Narvik Nappe Complex (Augland et al., 2014) and upper KNC (Kirkland et al., 2007a; Gasser et al., 2015).

### 6.6 An anticlockwise P-T path for Caledonian metamorphism in the RNC

The Vaddas, Kåfjord and Nordmannvik nappes all display an increase in pressure and/or decrease in temperature between pre-S2 or early-S2 high temperature, low pressure conditions and peak S2 conditions higher pressure, lower temperature conditions (Fig. 13A), recording a clear anticlockwise P-T path. Gabbro intrusion is syn-S1 during high temperature, low pressure conditions around ~440 Ma. Although the Vaddas and Kåfjord metasediments do not exhibit partial melting (with the exception locally around gabbro bodies), they also record this heating event. Anticlockwise metamorphism is recorded as the

Nordmannvik Nappe migmatites solidified, between 440-430 Ma, with an increase in pressure and decrease in temperature during Caledonian shearing (Figs. 13, 14). Although S2 pressures in the Kåfjord Nappe are not as high as those in the Nordmannvik and Vaddas nappes, the shift of the Kåfjord Nappe from an environment with a higher geotherm towards one with a lower geotherm (Fig. 13A) is also consistent with an anticlockwise P-T path, recording crustal thickening and subduction of pre-heated rocks (but beginning in a shallower crustal position).

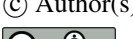



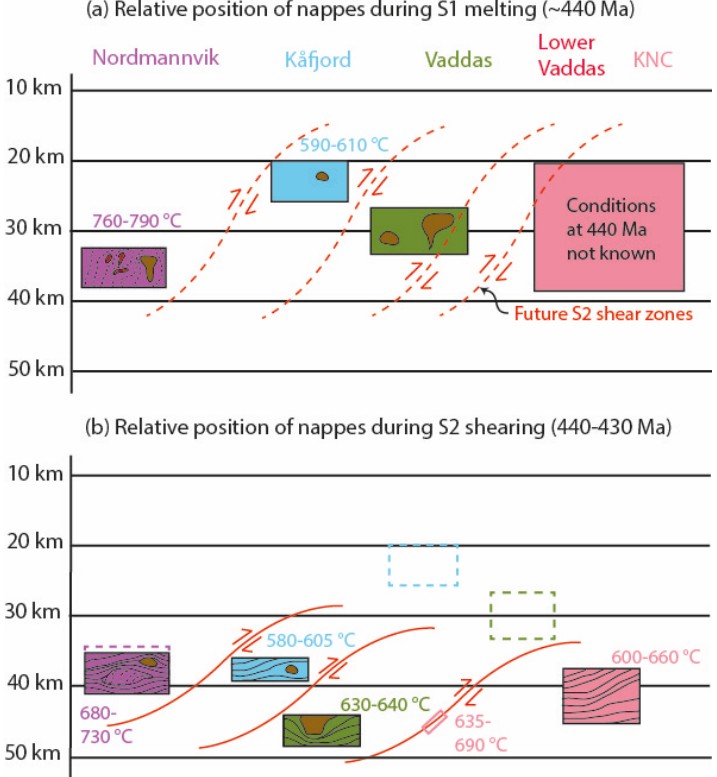

**Figure 14: Relative positions and temperatures of the RNC nappes and KNC during: (a) early Silurian heating (S1 melting and gabbro intrusion), and (b) S2 shearing, focussing on the sections through Arnøya and Uløya (lower Vaddas not present). S1 conditions are shown in dashed coloured boxes for comparison. In both Figs. (a) and (b) S1 is shown as a dashed black line, and S1 as a solid black line.**

## 7 Discussion: The tectonic evolution of the RNC

### 7.1 Depositional ages

In order to understand the age – metamorphic – deformation - relationships between the different nappes of the RNC and to reconstruct their tectonic evolution, the depositional ages of the metasedimentary rocks within the different nappes have to be known. The Vaddas and Kåfjord samples are metasedimentary rocks that so far have not shown any evidence for a pre-Caledonian metamorphic or deformation history. The Magerøy Nappe, considered equivalent to the Vaddas Nappe, was sedimented in the earliest Silurian based on fossil evidence and an age of 438 ± 4 Ma for volcaniclastic rocks, indicating deposition almost synchronously with emplacement of mafic-ultramafic and felsic magmas, and implying rapid burial of sediments (Andersen, 1981; Kirkland et al., 2005, 2016). Although no depositional ages for the Vaddas rocks associated with



the dated gabbro have yet been obtained, the presence of late Ordovician/early Silurian fossils at the base of the upper Vaddas metasediments south of the field area (e.g. Binns and Gayer, 1980) suggests that the upper Vaddas and possibly lower Kåfjord Nappes in our study area are late Ordovician or younger in age (Fig. 13B). The presence of sheared conglomerates at the base of the upper Vaddas Nappe (Figs. 3, 4D; Lindahl et al., 2005) suggests it was deposited on a continental basement; however it

is unclear what this basement was. The similarity of the metasediments in the lower Kåfjord Nappe and the upper Vaddas Nappe suggests that they might represent different stratigraphic and/or distal parts of the same late Ordovician basin.

The upper Kåfjord rocks, displaying evidence for an S1 foliation and an age of 440 Ma for anataxis (Dangla et al., 1978), are different from the lower Kåfjord metasediments, and probably represent strongly sheared Nordmannvik-derived rocks. We

therefore consider that the Kåfjord Nappe is probably comprised of sheared Nordmannvik-derived rocks in its upper part and sheared Vaddas-derived rocks in its lower part, however the pervasive nature of S2 shearing makes this difficult to define (Fig. 14B). The tectonostratigraphic position of the Nordmannvik paragneisses and their older zircon cores suggest they could also have been part of a more outboard basement to the Vaddas/Kåfjord metasediments (Fig. 15A-C). The protolith age of the Nordmannvik samples, however, is still unresolved. The presence of marble and calc-silicates suggests they were deposited as

shallow water sediments, probably on a continental shelf. The oldest inherited zircon cores (1800 – 1600 Ma) may represent detrital ages from the original source rock, and are typical intrusive ages in northern Baltica (e.g Larson and Berglund, 1992). The youngest inherited core age of 585 ± 8 Ma represents a maximum depositional age for these rocks (Figs. 11A, 13B). Farther to the southwest, the Narvik Nappe Complex, at the same tectonostratigraphic level, contains amphibolite-facies marbles with a chemostratigraphic depositional age of 610-590 Ma (Melezhik et al., 2014), indicating that the Nordmannvik

rocks could indeed represent Late Neoproterozoic deposits. Alternatively, if the sillimanite in garnet cores records some pre-Caledonian metamorphic event, then the inherited zircon core ages might not represent a detrital age but could record the timing of metamorphism associated with the sillimanite growth. More geochronology is required to better constrain the depositional age and pre-Caledonian history of Nordmannvik rocks.

**7.2 Early Silurian (~440 Ma) metamorphic and magmatic event**

Prior to the development of top-to-the-SE S2 shear zones, the KNC was the most inboard terrane relative to Baltica, with the lower Vaddas, upper Vaddas, Kåfjord and Nordmannvik nappes increasingly more outboard (Figs. 14, 15). Based on pressure estimates for conditions prior to S2 shearing, each nappe was also at different depths during this time, with the Kåfjord and Vaddas metasediments at mid- to lower crustal levels, respectively, and the Nordmannvik rocks at high temperatures and pressures (lower crustal level) during early Silurian heating (Fig. 14A). The garnet cores in the lower Kåfjord record a much

higher structural level than the Vaddas rocks. Together, the relative positions and relationships between the different nappes in the late Ordovician and Silurian suggests that the Vaddas and Kåfjord sediments were deposited in a continental basin (with KNC and/or Nordmannvik basement?) prior to early Silurian heating. The depth of gabbro intrusion requires that the Vaddas rocks were rapidly buried to ~26-34 km depth by ~440 Ma, suggesting either a very deep basin or thickened continental crust.





Widespread intrusion of gabbro across all three nappes at different crustal levels indicates extensive early Silurian magma underplating.

### 7.3 Similar Early Silurian metamorphic and magmatic events in the Scandinavian Caledonides

Early Silurian metamorphism and magmatism are recognized elsewhere in the Caledonides (e.g. Tucker et al., 1990; Gromet

et al., 1996; Vaasjoki and Sipilä, 2001; Andréasson et al., 2003; Kirkland et al., 2005; Corfu et al., 2011; Majka et al., 2012; Klonowska et al., 2013, 2017). Early Silurian migmatization in the Nordmannvik Nappe is similar to that observed in the Seve Nappe Complex (SNC), considered part of the outermost Baltica margin (continent-ocean transition zone). In the SNC, Ordovician high-pressure metamorphism (460 to 450 Ma) was followed by early Silurian (445-435 Ma) granulite facies metamorphism and magmatism (Gromet et al., 1996; Majka et al., 2012; Klonowska et al., 2013, 2017) at similar conditions

to those recorded in the Nordmannvik Nappe. However, no evidence for Ordovician high-pressure metamorphism has been observed in the RNC (sillimanite inclusions in garnet suggest high temperatures instead), and therefore, the tectonostratigraphic positions relative to Baltica of the RNC and the SNC seem to be quite different.

Intrusive rocks of early Silurian age are widely recognized along the length of the Caledonides. In northern Norway, they are

found in the Narvik Nappe Complex (mafic Rånå intrusion; $437 \pm 0.5$ Ma; Tucker et al., 1990), the Vaddas/Kalak Nappe (mafic Halti Igneous Complex; $434 \pm 5$ Ma and $438 \pm 5$ Ma; Vaasjoki and Sipilä, 2001; Andréasson et al., 2003), and granitic and gabbroic rocks on Magerøy (intruded between 440-435 Ma; Kirkland et al., 2005; Corfu et al., 2011). The ages and tectonostratigraphic relationships established in this work support a correlation of the Narvik Nappe Complex with the Nordmannvik Nappe (e.g. Augland et al., 2014). The Magerøy Nappe has been most closely associated with the Vaddas Nappe

(e.g. Andresen, 1988; Corfu et al., 2007). Both have similar lithologies and two-phase histories (D1 and D2 in the Magerøy Nappe; Andersen, 1981). Gabbro of the Honningsvåg igneous complex, which intrudes syn-D1, has a similar age as the Kågen gabbro in the Vaddas Nappe (Corfu et al., 2006). The Skarsvåg Nappe overlies the Magerøy Nappe, is comprised of migmatitic mica-schist and quarzites, and is intruded by granites synchronously with or after D1 at $436 \pm 1$ Ma and $435 \pm 2$ Ma (Andersen, 1984; Corfu et al., 2006). The similarity in lithology, ages, structures and tectonostratigraphy supports a correlation of the

Magerøy Nappe with the Vaddas Nappe and overlying Skarsvåg Nappe with the Nordmannvik Nappe.

Early Silurian mafic magmatism is also recognized south at Sulitjelma and extensively within the Trondheim Nappe Complex (Fig. 1A; Pedersen et al., 1992; Nilsen et al., 2007; Slagstad and Kirkland, 2018). The age window of this mafic magmatism is between 438 – 434 Ma (Slagstad and Kirkland, 2018), fitting with the ages determined for the Kågen gabbro and

migmatization in the Nordmannvik Nappe. Along the Caledonian orogen most occurrences are attributed to marginal basin settings, and typically considered as part of the Iapetus-derived rocks. In cases such as the TNC, oceanic rocks are recognized tectonostratigraphically below them (e.g. Andersen et al., 2012), in contrast to the RNC.



### 7.4 Large-scale tectonic models for the RNC

Several different tectonic scenarios have previously been proposed to explain the Ordovician high-pressure metamorphism and early Silurian mafic magmatism in the Scandinavian Caledonides (e.g. Pedersen et al., 1992; Northrup, 1997; Andréasson et al., 2003; Kirkland et al., 2005; Corfu et al., 2006; Slagstad and Kirkland, 2018). However, establishing tectonic models that

fit rocks in northern Norway is challenging due to the lack of typical arc rocks, and oceanic rocks marking a suture. The only oceanic rocks present belong to the Lyngsfjellet Nappe, which overlies the RNC. The marked discontinuity in metamorphic grade between the Nordmannvik (amphibolite-granulite facies) and Lyngsfjellet (greenschist-low amphibolite facies) nappes suggests that the Lyngsfjellet Nappe was not thrust in-sequence over the Nordmannvik Nappe, and that it did not reach the same depths as the RNC or KNC. The nature of the boundary between the Nordmannvik Nappe and overlying Lyngsfjellet

Nappe is still unclear, and it could also be a later detachment. Regardless, its position above the Nordmannvik Nappe means that it must have been somewhere outboard of the Nordmannvik Nappe rocks, with the overlying (Laurentia-derived?) Nakkedal and Tromsø nappes representing more exotic elements during the early Silurian (Fig. 15; Corfu et al., 2003; Janák et al., 2012; Augland et al., 2014). The tectonostratigraphic position of the RNC suggests it formed a part of the outer Baltica margin (e.g Fig. 15A).

Any tectonic model of the Caledonian evolution of the RNC in northern Norway needs to account for the tectonostratigraphy, Late Ordovician/early Silurian sedimentation and volcanism (Vaddas and Kåfjord), rapid burial and intrusion of tholeiitic gabbros (Vaddas) concurrent with migmatization in older, deeper rocks further outboard (Nordmannvik), followed by the onset of S2 shearing, pressure increase and cooling (anticlockwise P-T path), and the development of a lower crustal nappe stack.

Anticlockwise P-T paths are typically observed where an initial heating event is followed by burial or cooling (Wakabayashi, 2004). Nappe stacking itself is not considered to be capable of providing the amount of heat required to explain the estimated temperatures for the initial heating event (e.g. Johnson and Strachan, 2006). Ridge subduction (e.g. Northrup, 1997; Corfu et al., 2006) has been mentioned as a possible explanation for the apparent coeval deposition of volcaniclastic sediments and tholeiitic mafic intrusions and the relatively short time window of mafic magmatism. However, the effect of ridge subduction

on magmatism is usually very localized (e.g. Lomize and Luchitskaya, 2012). Ridge subduction therefore does not explain the widespread nature of the early Silurian mafic magmatism several hundreds of kilometers along strike of the Caledonian orogeny from the Magerøy Nappe in the north, southwards to Trondheim.

The widespread regional heating event is best explained by magmatic underplating of mantle melts. The gabbros with ~440
Ma intrusion ages (Kågen, Heia, Magerøy) are evidence for mantle melts at this time. Deposition of volcaniclastic sediments and tholeiitic mafic magmatism is often indicative of an extensional setting, such as a back-arc basin (e.g. Pedersen et al., 1992; Kirkland et al., 2005; Slagstad and Kirkland, 2018), and we consider this setting in several different tectonic scenarios. Considering the RNC in a back-arc position on the Baltica margin requires eastward-dipping subduction (e.g. Fig. 15A;



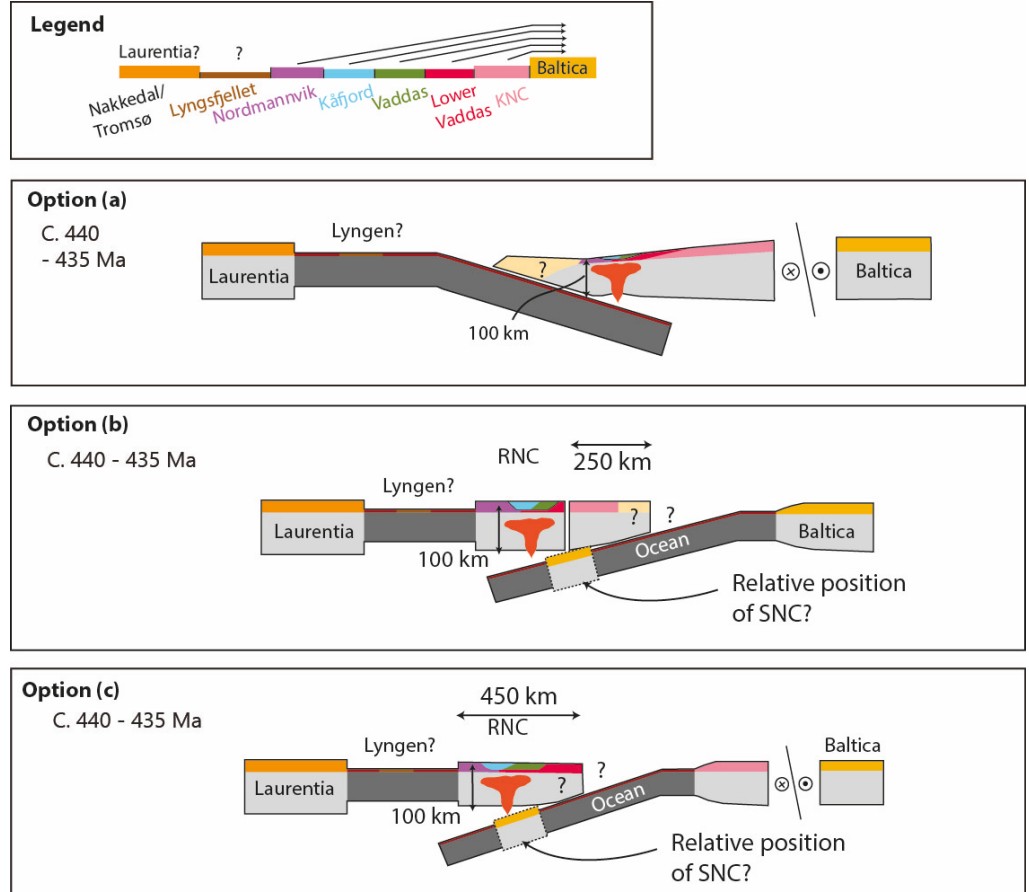

**Figure 15: Cartoons depicting some of the tectonic models discussed for the Caledonian nappes in the study area. The legend (top) shows the nappes in their relative paleogeographic positions prior to S2 nappe thrusting. Option (a) displays eastward-dipping subduction with the RNC as a back arc on an upper Baltica plate (e.g. Andréasson et al., 2003). Option (b) and (c) show westward-**
5    **dipping subduction with the RNC in a back-arc position. Option (b) has the KNC on the upper plate, whereas option (c) has the KNC on the down going plate.**

Andréasson et al., 2003). However, the KNC lacks the expected magmatism for this model, and there is no evidence for a

switch in subduction polarity immediately prior to collision. For a back-arc environment we therefore have to consider that

there is a missing suture below or within the RNC or KNC. Suture zones can be cryptic and difficult to identify in highly

10   deformed rocks, with oceanic rocks typically not present along the entire length of a suture (Dewey, 1977). Similar nappes

further south in the Caledonides do display evidence for underlying oceanic rocks (e.g. Andersen et al., 2012).



Models that consider the RNC as formed in an extensional back-arc setting on the upper plate during westward subduction (e.g. Fig. 15B, C; cf. Slagstad and Kirkland, 2018) provide a good explanation for Ordovician/early Silurian deposition of volcanic rocks and sediments (Vaddas & Kåfjord) and intrusion of mafic rocks (e.g., Kågen gabbro; large-scale magma underplating), and provide an early Silurian heat source for an anti-clockwise P-T path. The early collision and slab roll-back

model of Slagstad and Kirkland (2018) supports short-lived early Silurian deposition, mafic magmatism and anticlockwise P-T path with rapid switching from extension and heating to compressional S2 shearing (melt-filled S2 axial planes in the Nordmannvik Nappe). However, paleomagnetic data from the Honningsvåg igneous complex on Magerøy suggests it intruded the Magerøy Nappe rocks when they were ~1350 km north of Baltica, supporting a possible early Silurian separation between Baltica and the RNC (e.g. Fig. 15B, C; Torsvik et al., 1992; Corfu et al., 2006).

Figures 15B and C offer two different scenarios based on the paleogeographic position of the KNC. Both consider the Vaddas and Kåfjord having deposited in a back-arc basin on a continental basement. Option B places the KNC as an outboard microcontinent, while option C considers the KNC as part of the down-going plate. Although rocks equivalent to the SNC are unknown in northern Norway, in this scenario the edge of the Baltica continent or some continent-ocean transition zone would

have undergone subduction, consistent with Ordovician HP metamorphism in the SNC (Figs. 15A, B; e.g. Dallmeyer and Gee, 1986; Majka et al., 2014). Both scenarios require the presence arc rocks either tectonostratigraphically between the RNC and KNC or below the KNC, which we do not observe. It is possible that subduction was relatively short-lived, providing too little time for a proper arc to develop.

There are several problems that arise when considering the RNC rocks as having formed as part of a back-arc basin. The fairly high pressures of S1 migmatization in the Nordmannvik Nappe (kyanite-bearing; Fig. 13A) and deep-seated Kågen gabbro intrusion are not typical for an extensional environment, and intrusion of gabbro at 7-9 kbar into sediments in a back-arc basin requires a very deep basin, especially if it was short-lived (e.g. Slagstad and Kirkland, 2018). The P-T path for S2 shearing in the Vaddas and the near isothermal increase in pressure from early heating to S2 in the lower Kåfjord Nappe (Figs. 13A, 14)

indicate subduction of pre-heated Vaddas rocks (deeper) and Kåfjord (shallower) as part of the lower plate. Some models discussed here fit better with our observations from northern Norway than others. Eastward-dipping subduction (e.g. Fig. 15a; Andréasson et al., 2003) seems unlikely or, at least, evidence for it is lacking so far. It is likely that the KNC (or at least its upper part) formed a microcontinent on the upper plate (e.g. Fig. 15b), placing the suture below it. The RNC as a microcontinent on the upper plate (e.g. Fig. 15c) is a also a promising model, assuming that a yet unrecognized suture is present between the

KNC and RNC. Both of these models (Figs. 15b, c) are possible in a scenario where early Silurian heating and gabbro intrusion resulted from extensive magma underplating in a back-arc setting with a rapid switch to shortening and collision over ~10 million years. Discrepancies in the models could be due to uncertainties in observations (e.g. paleomagnetic datasets or P-T estimates), or due to oblique collision, introducing strike-slip aspects to the story, and a possible overprinted pre-collisional accretion history.



The temperature and pressure evolution of the Vaddas and Kåfjord nappes from 440 Ma to 430 Ma can be used to discuss large-scale nappe stacking. Both indicate a clear pressure increase while temperatures remained similar (Fig. 14). An internal shear zone between the Kåfjord and Vaddas units was active to produce the thrust stacking of these two units (the pressure

difference between them becomes more pronounced; Figs. 13A, 14). Both units combined show a pressure increase with respect to the Nordmannvik nappe, which maintained its depth position (Figs. 13A, 14). This relative pressure development corresponds to thrusting of the Nordmannvik Nappe over the Vaddas and Kåfjord nappes or to initial subduction of these units underneath the Nordmannvik. The Vaddas Nappe was at the same depth as the KNC during the onset of this thrusting, together with the shear zone between them. This scenario suggests an initiation of a subduction zone towards the West with

Kåfjord/Vaddas units at its leading edge or thrusting of the Nordmannvik nappe over Kåfjord/Vaddas during continental collision.

**8 Discussion: Nappe stacking in continental collision zones**

Even though the early Silurian tectonic history in northern Norway could have resulted from several different scenarios, the P-T evolution of the rocks indicates they formed a relatively hot and weak part of the mid- to lower crust (especially the

Nordmannvik Nappe) prior to Scandian nappe stacking (S2; Fig. 13A). This is in contrast with anorthositic and granulitic Baltica crust in the Western Gneiss Region (WGR) and Lofoten further south in Norway. These crustal segments were subducted to depths >100 km with little internal deformation (Engvik et al., 2000; Hacker et al., 2010; Froitzheim et al., 2016). The anorthositic and granulitic rocks are not fertile or require large amounts of $H_2O$ for partial melting. Conversely, the Nordmannvik rocks are fertile and show extensive partial melting. Caledonian deformation started under melt-present

conditions (~440 Ma), while pressures increased and temperatures decreased from S1 to S2 deformation, constituting a counterclockwise P-T path. Subsequently, pervasive deformation continued into solid state conditions (~430 Ma). This suggests that partial melts present in buried rocks played a key role in controlling the style of deformation in continental collision zones: rock strength is significantly decreased even with melt fractions as low as ~7%, and melt-bearing systems may facilitate strain localization (e.g. Rosenberg et al., 2005; Cavalcante et al., 2016). In northern Norway, the fertile Nordmannvik

rocks are significantly different in composition compared to the dry, strong Baltica basement of the WGR and Lofoten. Partial melting in the fertile subducted Nordmannvik rocks led to strain localization, which facilitated pervasive deformation, detachment from underlying substrate, and underthrusting by rocks lower in the nappe stack (shown by higher pressures in the Vaddas and KNC; Fig. 14B) and their incorporation into the nappe stack instead of being subducted. The partial melting is the critical weakening process in such a case. The response of the lithology to the metamorphic conditions exerts a key control on

the strength of the rocks in a continental collision zone and can decide whether rocks become subducted or not. In this way the structure, composition and rheology of the lower crust during continental collision determines the geometry and kinematics during subduction of continental crust.



## 9 Conclusions

The Caledonian evolution of petrology and deformation, and their timing in the RNC in northern Norway records a continuous sequence of events: 1) late Ordovician/early Silurian sedimentation and mafic volcanism in the Vaddas and Kåfjord nappes, 2) early high temperature partial melting in the Nordmannvik Nappe associated with a migmatitic fabric S1 and mafic

magmatism in the Vaddas, Kåfjord and Nordmannvik nappes at ~443-435 Ma, followed by 3) pervasive amphibolite-facies shearing, formation of the subhorizontal S2 fabric, and nappe stacking recorded at ~437-427 Ma. The transition from high-temperature-medium pressure S1 metamorphism to higher-pressure-lower-temperature S2 metamorphism results in an anticlockwise P-T path. Deformation between these two events is continuous, and S2 deformation began while the rocks were still hot. The tectonic model that best accounts for our observations considers early Silurian heating resulting from extensive

magma underplating in a back-arc system, followed rapidly (within 10 million years) by collision, crustal thickening and nappe stacking. Early Silurian high temperature metamorphism in the RNC (particularly the Nordmannvik Nappe) led to weakening in the rocks, promoting extensive deformation, dismembering of the continental crust, and pervasive S2 shearing during nappe-stacking in early collision. The thrusting of the Nordmannvik unit over Kåfjord/Vaddas units or the subduction of Kåfjord/Vaddas beneath Nordmannvik took place with the Nordmannvik/Kåfjord contact being a weak zone caused by partial

melting in the Nordmannvik unit.

### Data Availablity

Underlying data for this paper can be found in the supplement.

### Author Contribution

CF wrote most of the manuscript (with input from all co-authors), conducted ~90% of the field work, and gathered most of the
data (XRF, EMP, SEM, SIMS, and TIMS) with assistance and input from the co-authors. CF, HS, DG and PJ were involved in project design. Field data from Uløya, and XRF analysis, EMP analysis and earlier phase equilibrium modelling for the sample from there (UL248) was conducted as part of the MSc project of KK. HS, DG, PJ, JK and KK participated in some field work. JK and EKR gave advice on the phase equilibrium modelling, and EKR contributed previous knowledge of the rocks on Arnøya. FC assisted with sample preparation and TIMS analyses and contributed significantly to the discussion and
the design of Fig. 15.

### Competing interests

The authors declare that they have no conflict of interest.



## Acknowledgements

We thank L. Menegon for helpful discussions in the field. C. Tinguely is thanked for help with the EMPA at the University of Cape Town and K. Neufeld for help with the SEM at UiT The Arctic University of Norway. SIMS data were collected at the NORDSIM Laboratory, operated under an agreement between the research funding agencies of Denmark, Iceland,
Norway and Sweden, the Geological Survey of Finland and the Swedish Museum of Natural History. K. Lindén and G. Kenny assisted with the collection of SIMS data. We thank UiT The Arctic University of Norway and the Norwegian Research School for Dynamics and Evolution of Earth and Planets for supporting this work.

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
