# Peer review of "Anticlockwise metamorphic P-T paths and nappe stacking in the Reisa Nappe Complex in the Scandinavian Caledonides, northern Norway: evidence for weakening of lower continental crust before and during continental collision"

_Solid Earth, 2018_

## Referee Comment (RC1) · Dr. Andersen (Referee) · 10 Aug 2018

10. August 2018: Review of manuscript for Solid Earth by Carly Faber and co-authors: An anticlockwise metamorphic P-T path and nappe stacking in the Reisa Nappe Complex in the Scandinavian Caledonides, northern Norway: evidence for weakening of lower continental crust before and during continental collision

I have already commented on an early version of an 'early-version' manuscript (PhD

thesis chapter) from late last year (2017) when Carly Faber submitted and later defended it as part of her PhD at the University of Tromsø. We had a good discussion on a number of issues at the time, and I had a several comments, which as far as I can see (and remember) now have been dealt with in very good manner in this revised and improved version submitted for Solid Earth.

The paper is both well written and is almost without spelling and technical errors, it is an excellent revision from the earlier PhD manuscript. Figures (maps w/structural information and cross-section) and micrographs are well drafted and of good quality, respectively. Diagrams with geochronological results and metamorphic phase petrology modelling (Perplex) are well presented and of good quality. Interpretations of the metamorphic petrology are well supported by the data. There can be little doubt that there was a an Early-Silurian (∼440-438 Ma) high-temperature regional metamorphic event associated with local partial melting and abundant emplacement of mafic and some granitic igneous rocks, which was succeeded by an overall regional pressure increase associated with nappe amalgamation and emplacement.

The title of this paper is therefore well-supported by the data it presents.

In the abstract the term 'northern' Norway is used, this term includes the entire 3-county-region from Trøndelag to North Cape and all of Finnmark. I suggest they replace it with 'northernmost Troms in northern Norway' just to avoid the idea that all of Northern Norway is discussed here.

The regional geology and tectonic setting are also well-presented and the manuscript now also uses the well-established geological information from the Magerøy nappe in the type locality at Magerøya and in its lateral continuations in west Finnmark (Sørøya, Porsanger peninsula, Hjelmsøya) in a better way than the original drafts for the PhD. Correlations between this now discontinuous tectonic unit into the study area in northeasternmost Troms are better constrained.

The paper is well-organized with systematic descriptions of the lithological variations,

petrography, structure and metamorphic petrology of the individual tectonic units discussed. Each unit is well established and documented. Irrespective of future work, this is data that will remain important for any further investigations and tectonic interpretations of the region.

The geochronology and metamorphic petrography- and petrology is well-presented and integrated with the structural observations. This work is now of excellent quality. I have no comments to suggest improvements to this part of the work, and it is obvious to me that this work is now ready for publication in Solid Earth.

In the discussion and conclusion, the authors use the convincingly documented anticlockwise PT-time path during the Scandian assembly of the nappe stack in the study area to discuss large-scale tectonic model(s) that may explain their observations. This is done with care and the different models discussed are well treated. Previous ideas regarding the Kalak Nappe Complex (exotic or not?) are discussed and both models are also considered. Their conclusions are soundly based on the data presented and previous studies.

From this it is clear that the use of references (local both Caledonian and general regarding methods and models) are good. I foound that the Andersen (1981, Structure of the Magerøy nappe) paper referred to in txt is missing from ref-list, and there is a spelling error in ref Engvik et al. 200, Austrheim's name is with a single r (not a double 'r' as in the list).

In conclusion: I find that this is an excellent and important contribution to the understanding of the Caledonian infrastructure of northernmost Norway and that the paper should be published in 'Solid Earth' as presented by the authors in this manuscript.

Congratulations with the nice work, Torgeir B Andersen

---

## Referee Comment (RC2) · T. Nagel (Referee) · 23 Oct 2018

Dear Editor, Dear Authors,

here come my comments to the article "An anticlockwise metamorphic P-T path and nappe stacking in the Reisa Nappe Complex in the Scandinavian Caledonides, northern Norway: evidence for weakening of lower continental crust before and during continental collision" by Carly Faber and others. I enjoyed reading the article and apologize

for the late response. The manuscript is well and clearly written, it presents an enormous amount of data, and all figures are carefully done. Also in view of the special publication format (discussion article), I recommend publication at the present state - it is already out and I do not see why that status should be questioned. I do have a few comments and suggestions, but nothing that should be considered as a mandatory change before publication. I congratulate the authors to this nice work.

The manuscript presents field data leading to a convincing nappe correlation, structural data, petrological data combined with detailed thermodynamic modeling, and abundant U-Pb dating of zircon and titanite addressing magmatism and metamorphism. The fundamental conclusions that the Reisa Nappe Complex did not experience deep subduction prior to the Scandian event (like the Seve nappe in the same structural position further South) and was at high-temperature-low-pressure conditions at the onset of nappe stacking is in my view convincing. It represents an important and probably controversial contribution to unraveling the Caledonian orogen in northern Norway.

I have not done a review for this kind of article before. I do it in a common way and list some general remarks before going through the manuscript.

General comments

Consider "Paths" in title.

The abstract confronts the reader with a flood of data and I found it somewhat confusing. I would consider, not to squeeze in as much data as possible but to rather summarize the story.

The petrological data (especially microprobe data) is a little thin compared to the extended modeling exercise. For some samples, one would like to see element maps or, where there is a map, one would like to see more elements - at least in the supplement. Often, the text is phrased in a general way as if several samples were analyzed, but all that is presented is a couple of point analyses (e.g. sample AR71).

Data acquisition and modeling effort are quite asymmetric, and it is not clear to me why some samples were treated with so much more care than others. Sample UL248 is handled in a way accounting for garnet fractionation – why only this sample and not the likewise important sample R153? A lot of work is put into showing the anti-clockwise evolution in sample AR25b including the Raman probe analysis of sillimanite in garnet cores. Together with the diagram 10b, this results are maybe the most convincing case of an anti-clockwise PT path. And then, the authors discuss the results away: The inferred conditions might be wrong due to disequilibrium or even be pre-Caledonian. But why take the trouble of the Raman, if one has no explanation for a progressive evolution from sillimanite-stable to Ky + melt, anyway? One could think of explanations, why melting increased towards the Ky-field (e.g. flushing). On the other hand, the important sample A01 (in which titanite has been dated to capture metamorphism) is not treated petrologically at all.

A few comments on the thermodynamic modeling (fig. 7 and 10): I would extend the isopleths in figures 7 and 10 to all assemblages. One should be careful using Mn isopleths, especially in figure 7b. Garnet is the only Mn-containing phase (the only other considered phase, ilmenite, is not stable in discussed assemblages). That means, the amount of Mn in garnet reflects only the total amount of garnet. This becomes especially problematic, if garnet in a first assemblage is removed for subsequent modeling - except for an arbitrary amount (25%), which determines, how much Mn reenters the bulk. It would be easy to tune Mn-isopleths in figure 7b. Modern software actually allows calculating full or partial fractionation of minerals along defined PT paths. That way, one could also test if certain zoned minerals actually grow along a path. This would be required for interpreting zoning as growth zoning.

The authors interpret the upper part of the Kåfjord nappe (KN) as belonging to the Nordmannvik nappe (NN), the lower part as belonging to the Vaddas nappe (VN) (if I get it right). Why do they distinguish a KN at all? Also, the NN and the VN have completely different histories according to this study: While the VN was at the surface until lower Silurian times, the NN experienced high-temperature conditions maybe since Proterozoic times (if I get it right). So why are they put together in a RNC?

How did the Vaddas/Kåfjord nappe (or a part of it) get from the surface to the depth of intrusion (after the deposition of Silurian surface rocks)? If the nappe is seen to consist of different parts juxtaposed during nappe stacking, it should be stated more clearly.

How does one make a flat lying nappe pile of relatively thin nappes with large horizontal dimensions if the system was hot and shallow at the onset of thrusting? I would not expect a hot arc/backarc crust to deform in this style.

How was the temperature of metamorphism during intrusion of gabbro in the VD inferred? And what is it? There is no temperature in fig. 14A and no temperature mentioned in the text – but a thermal gradient (?). This could be the most convincing case of anti-clockwise PT-evolution. I would generally say a little more about the pressure determination from Getsinger et al. (2013).

Detailed comments

page 2/line 2 (2/2): Remove first sentence?!

2/16: "in Silurian to Devonian times"?

2/23: Maybe indicate, where the Magerøy nappe is located.

2/24: "units" instead of "rocks".

2/26-27: I think, a structural position can not be "paleogeographically unique". Maybe: "Its structural position corresponds to a distinct paleogeographic position." (that has to be unraveled)

4/14-15: Defining "Scandian" as "the main Caledonian deformation" is probably not a good idea and not the definition proposed/discussed in Corfu et al. (2014; "final stage of Caledonian orogeny", "after 430 Ma")

4/21: Should be made clear at the beginning of the sentence that this is about KNC (could be GNC).

4/23: There should be a few references at the end of this sentence.

4/30: "Outcrop" is not a verb. ("The RNC crops out...")

6/6" "axes of open folds"?

6/9" "black ones/ black lines".

12/18f: Reading this description I wondered why the KN is distinguished at all - the upper part seems to have affinity to NN, the lower to VN. If the defining characteristic is that the base cuts the gabbros, I would state this.

12/27f.: Does this not indicate that much of the deformation (S2) is post nappe stacking? Where is the nappe stacking event?

13/18-19: First, I thought that using the LOI for modeling the second stage would be a great idea, but if melt is part of the assemblage, it could have been considerably more H2O-rich and degassed during solidification, or not?

14/8: I see one questionable rutile-inclusion in the grt ruin in fig. 4C.

18/18ff: See above: documentation for these observations is pretty poor. The text suggests that several grains have been studied including the rim all around the grain. Of these several grains "very few" show asymmetric rims. What we get to see is one grain and the rim in one direction. Where is the other data?

19/4ff: Exhumation during shearing? (VN)

22/13: One should say a little about the sample material that was used for determining the bulk composition. Was is a mixture of melanosome and leukosome? In the following one should maybe emphasize that both stages in fig. 10A assume melt-present condition and thus no fractionation because of melt extraction.

22/16: "final"? (remove?)

24/8-9: This caveat hangs a little in the air – can stage 1 be anywhere in the stability field of the assemblage. Is there any way to constrain the range? From fig. 9A, I can not see evidence that the core was not affected by diffusion. Small changes in grt compositions make rather large PT variations in this assemblage, right?

25/3: Maybe remind reader of the structural position of sample (near boundary to underlying KN).

25/11: Remove "apparent"? I do not get the sense anyway – can sectioning create artificial bimodal grain-size distributions?

25/7-26: The text is hard to follow, both because of writing and convincing observations supporting the statements. The little data their is in fig.9 seem stretched to come to the generalizations. I cannot recognize two distinct generations and would like to see an element map. in Fig. 9C, I wonder how Xmg can vary from Xpy so little with Xalm=0.7-0.75.

28/3: Fluid or melt?

28/12: What kind of microstructural relationships would that be? The ones mentioned before?

31/21-22: A pressure-increase is not visible to me!

35/6: Mention also volcaniclastic rocks from Magerøy nappe.

35/33: What is meant by "thickened continental crust"? Is the Vaddas nappe composed of rocks that were at the surface and at 35 km depth before deformation? Would you call this "a nappe"? Should be discussed that nappe-internal strain is so intense.

36/11:What is significance of sillimanite? If it is pre-caledonian, the NN might have well been subducted at 450 Ma (and could correspond to the Seve nappe).

[Figure]

Figures Figure 1 does not have a very good technical quality in my pdf. Can it not be reproduced with a higher resolution?

Figure 4H. Are chl primary inclusions? chl seems to be consistently along cracks.

Figure 6. Other maps in supplement!? Is there really no map for AR153?

Figure 7. Why are isopleths of grt composition not seen through the entire grt stability range? I think this should be the case in order to see how unique the compositions are.

Table 3: I would not calculate Fe3+ in grt based on cation balancing! The spreadsheet obviously makes some assumptions (e.g. filling the T-site with Tschermak-Al that then has to be accounted for by Fe3+ in the A-site.) that are questionable and actually "drown" in measurement error. The scatter of Si around 3 with several measurements > 3 makes it problematic to interpret Si < 3 as Tschermak component. Besides - only one grt ends up having Fe3+.

Table 4: Some pl analyses rather poor. Xan = Ca/(Ca+Na+K) can be questionable, especially in sample AR153, where Ca seems low compared to Si, Al, and Na. However, a higher Xan yield even a better fit with the predicted PT range.

I hope the authors find my comments useful. I leave it to them to revise the manuscript or not.

Best Regards, Thorsten Nagel

---

## Author Comment (AC1) · 20 Nov 2018

We have made the following modifications based on T.B. Andersen's comments:

Comment 1

Comment: In the abstract the term 'northern' Norway is used, this term includes the entire 3- county-region from Trøndelag to North Cape and all of Finnmark. I suggest

they replace it with 'northernmost Troms in northern Norway' just to avoid the idea that all of Northern Norway is discussed here.

Change in the manuscript: "Northern Norway" has been changed to "northern Troms, Norway" in the abstract.

Comment 2:

Comment: From this it is clear that the use of references (local both Caledonian and general regarding methods and models) are good. I found that the Andersen (1981, Structure of the Magerøy nappe) paper referred to in txt is missing from ref-list, and there is a spelling error in ref Engvik et al. 2000, Austrheim's name is with a single r (not a double 'r' as in the list).

Changes in manuscript: 1. The Andersen, 1981 reference was added to the reference list. 2. The spelling of Austrheim's name has been corrected in the Engvik et al., 2000 reference.

We would like to thank T.B. Andersen for his helpful comments on and discussion of this manuscript.
* * *

---

## Author Comment (AC2) · 27 Nov 2018

Our responses and modifications based on T. Nagel's comments are as follows:

Comment 1. Consider "Paths" in title.

- Changes: Title changed to "Anticlockwise metamorphic P-T paths..."

2. The abstract confronts the reader with a flood of data and I found it somewhat

confusing. I would consider, not to squeeze in as much data as possible but to rather summarize the story.

- Changes: The abstract has been simplified and shortened and now reads as follows: This study investigates the tectonostratigraphy and metamorphic and tectonic evolution of the Caledonian Reisa Nappe Complex (RNC; from bottom to top, Vaddas, Kåfjord and Nordmannvik nappes) in northern Troms, Norway. Structural data, phase equilibrium modelling, and U-Pb zircon and titanite geochronology are used to constrain the timing and P-T conditions of deformation and metamorphism during nappe stacking that facilitated crustal thickening during continental collision. Five samples taken from different parts of the RNC reveal an anticlockwise P-T path attributed to the effects of early Silurian heating (S1) followed by thrusting (S2). At 439 Ma during S1 the Nordmannvik Nappe reached the highest metamorphic conditions at $\sim$760 – 790 °C and $\sim$9.4 – 11 kbar inducing kyanite-grade partial melting. At the same time the Kåfjord Nappe was at higher, colder, levels of the crust (590 – 610 °C, 5.5-6.8 kbar) and the Vaddas Nappe was intruded by gabbro at > 650 °C and 6-9 kbar. The subsequent S2 shearing occurred at increasing pressure and decreasing temperatures (680 – 730 °C, 9.5 – 10.9 kbar in the partially molten Nordmannvik Nappe, 580-605 °C and 9.2-10.1 kbar in the Kåfjord Nappe and 630-640 °C, 11.7-13 kbar in the Vaddas. Multistage titanite growth in the Nordmannvik Nappe records this evolution through S1 and S2 between about 440 and 427 Ma, while titanite growth along the lower RNC boundary records S2-shearing at 432 $\pm$ 6 Ma. It emerges that early Silurian heating ($\sim$440 Ma), probably resulted from large-scale magma underplating, initiated partial melting that weakened the lower crust, which facilitated dismembering of the crust into individual thrust slices (nappe units). This tectonic style contrasts with subduction of mechanically strong continental crust to great depths as seen in, e.g., the Western Gneiss Region further South.

3. The petrological data (especially microprobe data) is a little thin compared to the extended modeling exercise. For some samples, one would like to see element maps

or, where there is a map, one would like to see more elements - at least in the supplement. Often, the text is phrased in a general way as if several samples were analyzed, but all that is presented is a couple of point analyses (e.g. sample AR71). Data acquisition and modeling effort are quite asymmetric, and it is not clear to me why some samples were treated with so much more care than others. Sample UL248 is handled in a way accounting for garnet fractionation – why only this sample and not the likewise important sample AR153?

- Response: Sample UL248 was treated in this way because it was the only sample that showed field evidence indicating the likelihood that garnet cores might represent a pre-Caledonian event (i.e. that cores and rims represented two distinct metamorphic events). The sample was also the only one that displayed significant zoning with clear and sharp differences between the cores and the rims. We chose not to treat sample AR153 in the same way because there no evidence was present in the field structures to suggest that the cores might represent a different event. We have included more point analyses in a spreadsheet in the supplement (tables S4-S7) ) for the samples used for pseudosection modelling and consider those in tables 3 and 4 as representative of these analyses. Unfortunately microprobe mapping time was limited by funding, and we therefore chose to map UL248 using the probe as it showed the most significant zoning profile. Fe, Mn and Mg element maps for sample UL248 are also now included in the supplement (Figure S1).

4. A lot of work is put into showing the anti-clockwise evolution in sample AR25b including the Raman probe analysis of sillimanite in garnet cores. Together with the diagram 10b, these results are maybe the most convincing case of an anti-clockwise PT path. And then, the authors discuss the results away: The inferred conditions might be wrong due to disequilibrium or even be pre-Caledonian. But why take the trouble of the Raman, if one has no explanation for a progressive evolution from sillimanite-stable to Ky + melt, anyway? One could think of explanations, why melting increased towards the Ky-field (e.g. flushing).
- Response: We now consider melt removal as an explanation for the P-T modelling of sample AR25b.

- Changes: Page 30 line 26 now reads as follows "This is in contrast to the microstructures that suggest that substantial partial melting produced mainly kyanite. This scenario is only possible if earlier melt formed under sillimanite grade conditions was removed from the system and melting continued under kyanite-grade conditions. However, due to high rates of diffusion in garnet that prevail at the temperatures under consideration during the S1 partial melting and S2 shearing (e.g. Caddick et al., 2010), we have to consider other possibilities for the metamorphism related to the sillimanite inclusions; 1) subsolidus prograde..."

5. On the other hand, the important sample A01 (in which titanite has been dated to capture metamorphism) is not treated petrologically at all.

- Response: This rock is not treated petrologically as the other rocks because it is a calc-silicate with a completely different bulk composition to the modelled rocks (metapelites). Its bulk composition is not comparable and it does not contain as many useful indicator minerals for pseudosection modelling (compared to the metapelites). It was therefore not included as one of the main petrological samples. A metapelitic migmatite from nearby (sample AR25b) that did not contain titanite (due to its different bulk composition) was modelled as its bulk composition is more comparable to other samples and it better recorded S1 migmatization and S2 shearing on a macroscopic scale. Sample A01 is petrographically described in section 5.2. Given that the titanite dating method (TIMS) is not in situ it is difficult to determine if there is a link between age and different microstructural relationships.

6. A few comments on the thermodynamic modeling (fig. 7 and 10): I would extend the isopleths in figures 7 and 10 to all assemblages.

- Response: Although we agree that it can be useful to have all isopleths across the entirety of the pseudosection, it is not necessary for the P-T constraints. It is our

opinion that having multiple sets of isopleths (in some cases four different sets) across the entirety of each pseudosection makes them very difficult to read. And considering the length of the paper already, we have chosen not to add more figures. We therefore include the different isopleths for each pseudosection in the supplement as Figures S2-S4.

7. One should be careful using Mn isopleths, especially in figure 7b. Garnet is the only Mn-containing phase (the only other considered phase, ilmenite, is not stable in discussed assemblages). That means, the amount of Mn in garnet reflects only the total amount of garnet. This becomes especially problematic, if garnet in a first assemblage is removed for subsequent modeling - except for an arbitrary amount (25%), which determines, how much Mn re-enters the bulk. It would be easy to tune Mn-isopleths in figure 7b. Modern software actually allows calculating full or partial fractionation of minerals along defined PT paths. That way, one could also test if certain zoned minerals actually grow along a path. This would be required for interpreting zoning as growth zoning.

- Response: Garnet is not the only Mn-bearing phase being considered. Biotite is also an Mn-bearing phase that can exchange with garnet, and biotite is abundant in the sample. Mn is also considered in the biotite solid solution model used in the modelling (Bio(TCC)). Perple_X does therefore allow for fractionation along the specified path.

8. The authors interpret the upper part of the Kåfjord nappe (KN) as belonging to the Nordmannvik nappe (NN), the lower part as belonging to the Vaddas nappe (VN) (if I get it right). Why do they distinguish a KN at all? Also, the NN and the VN have completely different histories according to this study: While the VN was at the surface until lower Silurian times, the NN experienced high-temperature conditions maybe since Proterozoic times (if I get it right). So why are they put together in a RNC?

- Response: This is a good question, and I think that future studies are needed to establish this better. The Kåfjord Nappe, where it is defined in Kåfjord is significantly

thicker than on Arnøya. There it is still unclear what constitutes the upper and lower parts and the tectonostratigraphy in general is not well defined. We base this distinction mainly on field observations from Arnøya and Uløya, where the Kåfjord Nappe is significantly thinner. I think detrital age dating of the upper and lower parts of the Kåfjord Nappe and the Nordmannvik Nappe is required to determine how and why they might be related, and if they should belong to the same nappe complex. Given the size of this paper already, this is beyond the scope of the study.

9. How did the Vaddas/Kåfjord nappe (or a part of it) get from the surface to the depth of intrusion (after the deposition of Silurian surface rocks)? If the nappe is seen to consist of different parts juxtaposed during nappe stacking, it should be stated more clearly.

- Changes: - The intrusion of gabbro and its depth is now referred to in section 7.2 as follows: "The depth of gabbro intrusion requires that the Vaddas rocks were rapidly buried to ∼26-34 km depth by ∼440 Ma, suggesting either a very deep basin or thickened continental crust facilitated by initial collision."

- The different parts are described more clearly in section 3.3 (see comment 23).

- In terms of juxtaposition of the different parts, this is difficult to reconcile based on the pervasive nature of the deformation. The following was added to section 7.1 : "As some of the metamorphic temperatures are so high that potential basement cover relationships cannot be reconstructed, the distinction between different nappes becomes difficult and can only be made on the basis of age dating or lithological associations."

- Section 8 has been modified, including the following: "Additionally, S2 deformation is pervasive and internal nappe strain can be as high as strain along nappe boundaries. This is in contrast to typical Alpine-style nappes (e.g. Escher et al., 1993; Escher and Beaumont, 1997), where basement and cover relationships can be established on the basis of metamorphic grade or nappe dividers."

10. How does one make a flat lying nappe pile of relatively thin nappes with large horizontal dimensions if the system was hot and shallow at the onset of thrusting? I would not expect a hot arc/backarc crust to deform in this style.

- Response: This is a good question. The system was not that shallow at the onset of thrusting (kyanite-grade migmatization in the Nordmannvik Nappe indicates relatively significant depths). Pre-heating of the rocks during S1 metamorphism likely facilitated weakening, promoting dismembering of the slab. The shape of the nappes is probably a function of spatial variations, rheological strength and contrasts thereof. We suggest that these nappes are not formed in the same way Alpine Nappes are formed and now address this briefly in the introduction and discussion. While we do speculate, we think that this is a question that still needs to be answered with further work.

11. How was the temperature of metamorphism during intrusion of gabbro in the VD inferred? And what is it? There is no temperature in fig. 14A and no temperature mentioned in the text – but a thermal gradient (?). This could be the most convincing case of anti-clockwise PT-evolution. I would generally say a little more about the pressure determination from Getsinger et al. (2013).

- Response: The pressure of gabbro intrusion is well defined by (Getsinger et al., (2013) based on the presence of kyanite in pegmatites associated with the gabbro (the same pegmatites that were dated). The temperature is less well defined. It must be cooler than the temperature predicted by igneous biotite (900 °C) but warmer than the youngest metamorphism related to solid state shearing during nappe-stacking (650 ± 50 °C) estimated using HblPlag by Getsinger et al., (2013).

- Changes: Page 32, line 22 now reads "Getsinger et al., (2013) established the P-T range based on the presence of kyanite in syn-tectonic pegmatites related to gabbro intrusion, and on the composition of igneous biotite and metamorphic hornblende and plagioclase. The temperature range for intrusion is rather large (650-900 ± 50 °C), but the pressures are rather well constrained by the presence of zoisite at lower temperature shearing and kyanite in pegmatites (700-900 MPa). The data indicates a depth of intrusion of âĹij26-34 km."

12. page 2/line 2 (2/2): Remove first sentence?!

- Changes: "Continental collision is one of the most important processes in plate tectonics." removed as first sentence.

13. 2/16: "in Silurian to Devonian times"?

- Changed to "in Silurian to Devonian times" 14. 2/23: Maybe indicate, where the Magerøy nappe is located.

- Changes: "located in northern Finnmark" added

15. 2/24: "units" instead of "rocks".

- Changes: "rocks" changed to "units"

16. 2/26-27: I think, a structural position can not be "paleogeographically unique". Maybe: "Its structural position corresponds to a distinct paleogeographic position." (that has to be unraveled)

- Changed to "places it in a distinctive paleogeographic position".

17. 4/14-15: Defining "Scandian" as "the main Caledonian deformation" is probably not a good idea and not the definition proposed/discussed in Corfu et al. (2014; "final stage of Caledonian orogeny", "after 430 Ma")

- Response: Corfu et al., (2014) defines "Scandian" as follows: "The Scandian phase of the Caledonian orogeny represents the main continent-continent collision between Baltica-Avalonia and Laurentia after the closure of the Iapetus Ocean." This is dated by the age of the youngest marine deposits identified along the suture, which are 430 Ma (Brekke & Solberg, 1987; Corfu et al., 2006).

- Page 4 line 17 changed to "The predominant Caledonian deformation, associated

with continental collision (often referred to as Scandian. . ."

18. 4/21: Should be made clear at the beginning of the sentence that this is about KNC (could be GNC).

- Changed to "recent evidence shows that the sedimentary cover in the KNC had already been deposited".

19. 4/23: There should be a few references at the end of this sentence.

- The following references were added: (Daly et al., 1991; Kirkland et al., 2006; Corfu et al., 2007, 2011).

20. 4/30: "Outcrop" is not a verb. ("The RNC crops out...")

- Changed to "The RNC crops out east of Lyngen. . ."

21. 6/6" "axes of open folds"?

- Changed to "axes of open folds" in caption of Figure 2.

22. 6/9" "black ones/ black lines".

- Changed to "black ones" in caption of Figure 2.

23. 12/18f: Reading this description I wondered why the KN is distinguished at all – the upper part seems to have affinity to NN, the lower to VN. If the defining characteristic is that the base cuts the gabbros, I would state this.

- Changed to "Although the lower part of the Kåfjord Nappe in the field area is comprised of similar metasediments to the Vaddas Nappe (suggesting they may be related), it is defined as its own nappe because its base cuts he upper part of the gabbro in the Vaddas Nappe on Arnøya. All nappes show significant thickness variations, which partly are due to the undeformed Kågen and Kvænangen gabbros that are large boudins. Erosion has removed all units above them, e.g. the Kåfjord and Nordmanvik nappes (Fig. 3). The entire Vaddas Nappe, as it is currently defined, shows thicknesses

of 100-1500 m with its lower part wedging out towards the west and north."

24. 12/27f.: Does this not indicate that much of the deformation (S2) is post nappe stacking? Where is the nappe stacking event?

- Response: The nappe stacking event is associated with the pervasive S2 structures with consistent top-to-SE shear sense, which is considered to indicate nappe transport during thrusting. The nappe cores and nappe boundaries show no difference in orientation of structures or kinematic indicators, and nappe boundaries themselves are diffuse. These nappes cannot be thought of as a traditional Alpine-style nappe stack with discrete deformation along nappe boundaries. Deformation in the nappes described here is considered to have been facilitated rather by large scale ductile flow, resulting in pervasive deformation during nappe juxtaposition.

25. 13/18-19: First, I thought that using the LOI for modeling the second stage would be a great idea, but if melt is part of the assemblage, it could have been considerably more H2O-rich and degassed during solidification, or not?

- Response: In almost all cases (sample AR25b is the exception) the second stage (S2 foliation) is metamorphism during solid-state deformation, so there is no melt to degass in these cases. For sample AR25b LOI was used as a proxy for water content during S2 deformation (which initiated in the rock while it was in a molten state and so there potentially more water around that indicated by the LOI. We consider LOI as a minimum water content in the model.

26. 14/8: I see one questionable rutile-inclusion in the grt ruin in fig. 4C.

- Response: The quality of the image is compromised by the size restriction of the manuscript during submission. The rutile inclusion is clearer on a higher resolution image.

27. 18/18ff: See above: documentation for these observations is pretty poor. The text suggests that several grains have been studied including the rim all around the grain.

Of these several grains "very few" show asymmetric rims. What we get to see is one grain and the rim in one direction. Where is the other data?

- Response: Additional data, including these garnet analyses, has been included in the supplement in tables S4-S7. The data shows that the majority of the rim analyses do not show the same zoning as the thin asymmetric rim recorded by the transect and by the garnet analysis in Table 3.

28. 19/4ff: Exhumation during shearing? (VN)

- Response: Yes, this is discussed later in the manuscript in section 7.4.

29. 22/13: One should say a little about the sample material that was used for determining the bulk composition. Was is a mixture of melanosome and leukosome? In the following one should maybe emphasize that both stages in fig. 10A assume melt-present condition and thus no fractionation because of melt extraction.

- Changes: The following was added at the beginning of section 4.5.1; "The sample material used to determine bulk composition was trimmed of as much leucosome as possible so that it contained ∼5-10 vol% leucocratic material."

30. 22/16: "final"? (remove?)

- Changed to "Water content used in the pseudosection. . ."

31. 24/8-9: This caveat hangs a little in the air – can stage 1 be anywhere in the stability field of the assemblage. Is there any way to constrain the range? From fig. 9A, I can not see evidence that the core was not affected by diffusion. Small changes in grt compositions make rather large PT variations in this assemblage, right?

- Changes: The following was added "Considering this, and given the presence of sillimanite inclusions in garnet, it is possible that garnet core conditions could have been up to ∼2-3 kbar lower pressure and ∼50 °C hotter than the predicted estimate based on their current composition."

32. 25/3: Maybe remind reader of the structural position of sample (near boundary to underlying KN).

- Changed to "Sample AR26 (Figs. 2, 3; Table 1), which comes from just above the Nordmannvik-Kåfjord boundary, is a fine-grained..."

33. 25/11: Remove "apparent"? I do not get the sense anyway – can sectioning create artificial bimodal grain-size distributions?

- Changed to "The bimodal garnet grain size is also observed in hand specimen and is therefore not an artifact of sectioning."

34. 25/7-26: The text is hard to follow, both because of writing and convincing observations supporting the statements. The little data there is in fig.9 seem stretched to come to the generalizations. I cannot recognize two distinct generations and would like to see an element map. in Fig. 9C.

- Response: They are not two distinct generations, rather two populations of garnet (large and small grain size) with the smaller grain size garnets more affected by diffusion than the larger population. It is unclear what originally caused the two different populations. They have slightly different core compositions (reflected better in Table 3 than in the profiles), and the smaller garnets display more pronounced diffusion zoning in their profiles due to their smaller size. This has been made clearer by the following modification of the text (below). The garnet profile suggests zoning is not pronounced enough that it would show anything significant in a map.

- The paragraph was reworded to make the above points clearer as follows: "The larger garnets (Fig. 9B) display a slight difference in composition relative to the smaller ones (Fig. 9C). Cores in the larger garnets have slightly lower Sps and Grs contents (Sps1-1.5, Grs4-5) than cores in the smaller garnets (Sps2.5-3.5, Grs5-6). Almandine content in the larger garnet cores (Alm71-76) is higher than in the smaller garnet cores (Alm65-72; Table 3). The larger garnets display a relatively flat profile with thin rims, while the

smaller garnets exhibit more pronounce zoning (Fig. 9B, C). Inclusion-poor cores of the larger garnets appear microstructurally distinct from inclusion-rich rims, however Grs is the only end-member that reflects this difference (Fig. 9B, grey box). The larger garnets have a thin 10-20 $\mu$m-wide rim (Fig. 9B; dark grey box) with lower XMg values and Grs contents than cores, whereas smaller garnets have a similar zoning profile, but lack the large flat cores. Based on the similarity in rim compositions, and the lack of a chemical zoning following the microstructure (inclusion free cores in large garnets), it appears that the slight compositional variations in both the large and small garnets probably resulted from diffusion during high-grade metamorphism. The smaller garnets are probably more completely re-equilibrated than the larger ones. The two generations of biotite have different compositions."

35. 28/3: Fluid or melt?

- Response: Melt. Hoskin and Schaltegger (2003) suggest the following: "The origin of oscillatory zoning in zircon probably involves a kinetic feedback mechanism operating at the crystal/melt interface."

36. 28/12: What kind of microstructural relationships would that be? The ones mentioned before?

- Changed to "These microstructural relationships..." as the sentence refers to the previous two sentences describing microstructural relationships.

37. 31/21-22: A pressure-increase is not visible to me!

- Response: Fair enough. It depends on how much error is in the estimates. This does not change that it shows an anticlockwise P-T path, though.

- Changed to "This progression represents an anticlockwise P-T path (mainly with temperature decreasing) from S1 to S2 conditions (Fig. 13A)."

38. 35/6: Mention also volcaniclastic rocks from Magerøy nappe

- Changed to "The similarity of the metasediments in the lower Kåfjord Nappe, upper Vaddas Nappe and early Silurian volcaniclastic rocks of the Magerøy Nappe suggests that they might represent different stratigraphic and/or distal parts of the same late Ordovician-early Silurian basin."

39. 35/33: What is meant by "thickened continental crust"? Is the Vaddas nappe composed of rocks that were at the surface and at 35 km depth before deformation? Would you call this "a nappe"? Should be discussed that nappe-internal strain is so intense.

- Response: Both the introduction and section 8 have been modified to better discuss the difference in nature between the lower crustal Caledonian Nappes, with their internal nappe strain, and more typical Alpine-style Nappes. For "thickened crust" see comment 23.

- The following was added to the introduction: "The nappe concept of large scale thrust units was developed in the Alps (Bertrand 1884, Schardt 1893). The nappe units were defined on the basis of stratigraphy (in the sediments), or, in higher grade metamorphic "basement" units, by nappe dividers of cover sediments between these (e.g., Pfiffner 2014). When the metamorphic temperatures during thrusting are high in all units, the distinction between basement and cover units becomes virtually impossible. Under such high grade conditions, the thrust units typically are thin parallel rock slices of variable extent, and the distinction of individual nappe units can only be made on the basis of metamorphic grade or age of metamorphism/deformation. This is the situation in the western part of the northern Caledonides, where only high grade rocks of the deep part of the orogen form the nappe stack."

- The following was added to Section 8: "Additionally, S2 deformation is pervasive and internal nappe strain can be as high as strain along nappe boundaries. This is in contrast to typical Alpine-style nappes (e.g. Escher et al., 1993; Escher and Beaumont, 1997), where basement and cover relationships can be established on the basis of

metamorphic grade or nappe dividers."

- The following has been added to section 8: "Partial melting in the fertile subducted Nordmannvik rocks led to strain localization and decoupling of nappe units from the crustal part of the lithosphere, and facilitated pervasive deformation (even in the internal parts of the nappes). Continental crust units are underthrusted and stacked as nappes (shown by increasing pressures in the Vaddas and KNC; Fig. 14B) instead of being subducted. This process destroys the integrity of the lithospheric slab and produces thin crustal nappe slivers."

40. 36/11: What is significance of sillimanite? If it is pre-caledonian, the NN might have well been subducted at 450 Ma (and could correspond to the Seve nappe).

- Response: Yes, it could have formed in the same depositional environment as the Seve Nappe, however we do not see the same record of HP Ordovician subduction in the RNC rocks. The Seve Nappe rocks do not contain sillimanite inclusions in garnet either. Even if the sillimanite inclusions in garnet in the Nordmannvik Nappe are pre-Ordovician, the lack of evidence for Ordovician HP metamorphism in the RNC still suggests a different subduction history to the SNC.

- Changed to: "However, no evidence for Ordovician high-pressure metamorphism has been observed in the RNC (sillimanite inclusions in garnet suggest pre-Silurian high temperatures instead). The current lack of evidence for Ordovician high-pressure metamorphism in the RNC indicates that the RNC and SNC may have been in different tectonic positions relative to Baltica at that time."

Figures

41. Figure 1 does not have a very good technical quality in my pdf. Can it not be reproduced with a higher resolution?

- Response: The resolution was restricted due to size constraints for submission of the manuscript. The final version is of higher quality.

42. Figure 4H. Are chl primary inclusions? chl seems to be consistently along cracks.

- Response: Yes, they are primary. Inclusions in other grains are not along cracks. This image best shows the core-rim structure present in some garnets. There is also a chlorite inclusion indicated in the core of the garnet in Fig. 6c that is clearly not along a crack.

43. Figure 6. Other maps in supplement!? Is there really no map for AR153?

- Response: We had limited funds for microprobe mapping. Garnet transects indicate that zoning in garnets in AR153 is gradual, as do multiple spot analyses. The other maps for sample UL248 are now included in the supplement as Figure S1 and more garnet spot analyses are included in the supplement as table S4.

44. Figure 7. Why are isopleths of grt composition not seen through the entire grt stability range? I think this should be the case in order to see how unique the compositions are.

- Response: We have included the different isopleths over the entire range for each pseudosection in the supplement as Figures S2-S4.

45. Table 3: I would not calculate Fe3+ in grt based on cation balancing! The spreadsheet obviously makes some assumptions (e.g. filling the T-site with Tschermak-Al that then has to be accounted for by Fe3+ in the A-site.) that are questionable and actually "drown" in measurement error. The scatter of Si around 3 with several measurements > 3 makes it problematic to interpret Si < 3 as Tschermak component. Besides – only one grt ends up having Fe3+.

- Response: We chose to calculate Fe3+ because it is considered in the solid solution garnet model used in the pseudosection modelling.

46. Table 4: Some pl analyses rather poor. Xan = Ca/(Ca+Na+K) can be questionable, especially in sample AR153, where Ca seems low compared to Si, Al, and Na. However, a higher Xan yield even a better fit with the predicted PT range.

- Response: We agree. Unfortunately, these are the best analyses we have for plagioclase. However, it makes little difference to the estimates as they are constrained using garnet composition and mineral assemblage in all cases, with plagioclase compositional isopleths plotted for confirmation only.

References

1. Brekke, H. & Solberg, P. O. 1987. The geology of Atløy, Sunnfjord, western Norway. Norges geologiske undersokelse Bulletin, 410, 73–94. 2. Corfu, F., Torsvik, T. H., Andersen, T. B., Ashwal, L. D., Ramsay, D. M. & Roberts, R. J. 2006. Early Silurian mafic-ultramafic and granitic plutonism in contemporaneous flysch, Mageroy, northern Norway: U–Pb ages and regional significance. Journal of the Geological Society, London, 163, 291–301. 3. Getsinger, A.J., Hirth, G., Stünitz, H., and Georgen, E.T.: Influence of water on rheology and strain localization in the lower continental crust, Geochem. Geophys., 14, 2247-2264, doi.org/10.1002/ggge.20148, 2013. 4. Hoskin, P.W.O., and Schaltegger, U.: The composition of zircon and igneous and metamorphic petrogenesis, Rev. Mineral. Geochem., 53, 27-62, doi.org/10.2113/0530027, 2003.

The additional files referred to above are attached as part of the supplementary material.

We would like to thank T. Nagel for his helpful comments and discussion of the manuscript.

Please also note the supplement to this comment:
https://www.solid-earth-discuss.net/se-2018-74/se-2018-74-AC2-supplement.zip

---

## Author Response (AR2)

**Comments and changes to manuscript**

Comment 1

(i)       when referring to a regional tectonic event you are referring to deformation/metamorphism stage/s that include(s) not only tectonic foliation (S tectonites) but likely also a lineation (S-L tectonites). Maybe more appropriate to replace "S" with "D" when dealing with tectonic episodes;

Response and changes:

We originally chose not to assign $S_1$ and $S_2$ to $D_1$ and $D_2$ as multiple events are defined in the underlying Kalak Nappe Complex, and these events are not referred to in the same way in different papers. We therefore chose to stay out of the confusion. We have now changed $S_1$ and $S_2$ to $D_1$ and $D_2$ in the manuscript, figure captions and on figures, where appropriate.

To avoid confusion with events in the Kalak Nappe Complex, we have added the following to section 3.2: "We associate $S_1$ in the RNC with $D_1$ and $S_2$ and $L_2$ with $D_2$. $S_1$ in the KNC may be older than the $S_1$ in the RNC (pre-$D_1$)." And to section 6.1: "Our results indicate that $S_1$ in the KNC is significantly older than $S_1$ in the RNC. The $S_1$-forming event in the KNC was termed $D_1$ by Gasser et al., (2015). It is not the same $D_1$ as we describe here in the RNC. In this paper we consider metamorphism in the KNC as pre-$D_1$ relative to the RNC. " We also added $D_1$ and $D_2$ labels to Figure 13A for extra clarification.

Comment 2

(ii)       the error bars for ages and P-T estimates should be taken into consideration when presenting these data in the text (abstract, main text and Conclusions). In the text is written that errors for P-T are +/- 50 °C and +/- 1 kbar and this should be taken into consideration when presenting the thermo-baric conditions. The error bars for ages vary from +/- 1 to +/- 6 Ma. This is consistent with what presented in Figure 15, where the tectonic reconstruction is presented for time lapses of 5 Ma. Text should be revised accordingly (see also specific comments below);

Response and changes:

We have corrected all reference to ages and added error bars. When ages are referred to as a general age (without an error bar) they are also referred to using ca.

Usually with phase equilibria modelling constraints using pseudosections it is not possible to calculate individual errors related to individual P-T estimates, and errors related to geological uncertainty are not typically reported. We state that maximum possible errors are ± 50 C and ± 1 kbar in the text, which is based on statistical analysis of geological uncertainty in a recent paper (Palin et al. 2016), for the purposes of comparing estimates for different samples. These are maximum errors considered to be inherent in all P-T estimates obtained using phase equilibria modelling. We therefore added the following: "Although it is not possible to constrain uncertainties for individual P-T estimates, considering maximum possible errors based on geological uncertainty of $\pm$ 50 °C and ±1 kbar (e.g. Palin et al.,

2016),…". In the abstract, discussion and conclusions the estimated P-T conditions are rounded and now referred to using ca. to remove ambiguity.

Comment 3

(iii)     editing: "S1" and "S2" with "1" and "2" as subscript.

Changes:

This has been done throughout the manuscript, figure captions and figure labels.

Below some specific points that regards only the Abstract and Conclusions, but these issues should be fixed in the entire manuscript;

Abstract

5. line 19: "S1", "1" as subscript, here and throughtout the ms. + add "foliation" after "S1"

6. line 19: "S2", "2" as subscript, here and throughtout the ms. + add "foliation" after "S2"

Changes: This has been changed throughout the entire manusctript.

7. line 19: "At 439 Ma", please indicate the error bar for the age or better to change with "At ca. 440 Ma"

Changed to: At ca. 439 Ma

8. line 20: "9-11.4 kbar", decimal approximation for pressure estimates; which the error bar? maybe better to substitute with " 9-11 kbar"

Changed to: 9 – 11 kbar.

9. line 21 (same as above, the P-T range), "(590 – 610 °C, 5.5-6.8 kbar)", maybe better "ca. 600 °C and 6-7 kbar".

Changed to: ca. 600 °C and 6-7 kbar

10. line 23 & 24: please correct the provided P-T range

Estimated values are rounded, and changed to the following: "ca. 600 °C and 9-10 kbar in the Kåfjord Nappe and ca. 640 ºC, 12-13  kbar in the Vaddas Nappe"

11. line 25: "440 and 427 Ma", which the error bar for the ages? "440-425 Ma"

[revised manuscript text omitted]